# RILe: Reinforced Imitation Learning

## Abstract

Acquiring complex behaviors is essential for artificially intelligent agents, yet learning these behaviors in high-dimensional settings, like robotics, poses a significant challenge due to the vast search space. There are three main approaches that address this challenge: 1. Reinforcement learning (RL) defines a reward function, which requires extensive manual effort, 2. Inverse reinforcement learning (IRL) uncovers reward functions from expert demonstrations but relies on a computationally expensive iterative process, and 3. Imitation learning (IL) directly compares an agent's actions with expert demonstrations; however, in high-dimensional environments, such binary comparisons often offer insufficient feedback for effective learning. To address the limitations of existing methods, we introduce RILe (Reinforced Imitation Learning), a framework that learns a dense reward function efficiently and achieves strong performance in high-dimensional tasks. Building on prior methods, RILe combines the granular reward function learning of IRL and computational efficiency of IL. Specifically, RILe introduces a novel trainer–student framework: the trainer distills an adaptive reward function, and the student uses this reward signal to imitate expert behaviors. Uniquely, the trainer is a reinforcement learning agent that learns a policy for generating rewards. The trainer is trained to select optimal reward signals by distilling signals from a discriminator that judges the student's proximity to expert behavior. We evaluate RILe on general reinforcement learning benchmarks and robotic locomotion tasks, where RILe achieves state-of-the-art performance.

## 1 Introduction

Learning complex behaviors is critical for the advancement of artificially intelligent agents in many fields such as robotics. Over the years, reinforcement learning (RL) has emerged as a powerful framework for teaching agents to perform sophisticated tasks, yet it often requires extensive manual reward function design. This is both time-consuming and error-prone. There are two broad classes of methods that address the reward engineering problem: inverse reinforcement learning and imitation learning.

First, Inverse Reinforcement Learning (IRL) (Ng & Russell, 2000; Ziebart et al., 2008) infers the reward function from expert demonstrations, reducing the burden of manual reward engineering. Although IRL yields a reward function capable of providing nuanced feedback at different stages of learning, the iterative process of learning such a reward function is computationally expensive (Zheng et al., 2022), especially in high-dimensional environments with a large state-action space. To address this inefficiency, Adversarial Inverse Reinforcement Learning (AIRL) (Fu et al., 2018) learns the reward function within a binary classifier (discriminator), which is trained to distinguish expert data from those produced by the learning agent. However, the learned reward function in AIRL is optimized as a component of a binary classifier, constraining it to serve the primary objective of classification rather than being optimized to provide the most effective guidance for the learning agent.

Second, Imitation learning (IL) bypasses explicit reward design by directly comparing learned behaviors to expert demonstrations. Traditional IL approaches such as Behavioral Cloning (BC) (Bain & Sammut, 1995) match learned actions with expert demonstrations directly, requiring a substantial amount of expert data in high-dimensional tasks. Adversarial Imitation Learning (AIL), such as GAIL (Ho & Ermon, 2016), employs a binary classifier (discriminator) that distinguishes expert data from those produced by the learning agent, and uses it as the reward function. Using the direct

output of this classifier as a reward signal provides limited information. This sparse signal indicates whether a behavior is expert-like or not, but often fails to provide a useful guidance when the agent's behavior is far from the expert demonstrations, a common scenario in high-dimensional tasks where most behaviors are simply classified as non-expert.

Real-life learning scenarios suggest a different approach that does not rely on a judge to guide the learning process: think of parents and children, or a pet owner and their dog. The *teacher* also refines how they teach as the student progresses. Each success or failure in the student's understanding shapes the teacher's approach: lessons learned from suboptimal student behaviors ultimately yield better trainers, which, in turn, guide the student more effectively and result in better students. While Adversarial Imitation Learning (AIL) updates a policy and a discriminator simultaneously, their relationship is fundamentally adversarial. The discriminator's objective is to minimize its classification error, an objective that is better achieved when the student agent produces easily distinguishable, non-expert behaviors. Therefore, the discriminator's objective is fundamentally misaligned with that of a teacher: it is rewarded for identifying failure, not for creating a path to success.

Inspired by these insights, we propose Reinforced Imitation Learning (RILe). RILe combines the reward learning benefits of IRL with the computational efficiency of AIL (Fig. 1-(d)). RILe is a novel *trainer-student* system that is composed of:

- **Student Agent:** Learns a policy to imitate expert demonstrations using reinforcement learning.
- **Trainer Agent:** Simultaneously learns a reward function using reinforcement learning, leveraging an adversarial discriminator for continuous feedback on student performance.

RILe's trainer continuously updates the reward function in tandem with the student's policy updates, whereas IRL refines its reward function only after training a policy to convergence on the current reward function. Specifically, the trainer queries a discriminator to measure how expert-like the student's behavior is, then optimizes the reward function based on that feedback, without waiting for the policy to converge. Similar to IRL, RILe offers nuanced reward learning, while avoiding IRL's heavy computational loop. Our contributions are two-fold:

1. **Efficient Reward-Function Learning via RL**: We introduce a reinforcement-learning-based approach for training a reward function simultaneously with the policy. This avoids IRL's repeated policy re-training and learns the reward function efficiently.
2. **Dynamic Reward Customization:** RILe offers context-sensitive guidance that adapts to the student's evolving skill level, because the trainer agent updates the reward function as student evolves. This dynamic reward shaping is valuable in high-dimensional tasks, where the learning agent requires different forms of rewards in different stages of the training.

We evaluate RILe in comparison to state-of-the-art methods in AIL, IRL, and AIRL: GCL (Finn et al., 2016), REIRL (Boularias et al., 2011), GAIL (Ho & Ermon, 2016) AIRL (Fu et al., 2018), GAIfO (Torabi et al., 2018b), BCO (Torabi et al., 2018a), IQ-Learn (Garg et al., 2021) and DRAIL (Lai et al., 2024). Our experiments span six studies: (1) Comparing different trainer-discriminator relationships in RILe, (2) Empirically analyzing how RILe's reward-learning differs from baselines, (3) Comparing the computational cost of RILe with baselines, (4) Assessing RILe's performance in both low- and high-dimensional continuous-control problems, and (5) Analyzing the effect of using a more advanced discriminator in RILe. Our results show RILe achieves state-of-the-art performance, particularly in high-dimensional environments.

## 2 RELATED WORK

We review research on learning from expert demonstrations, focusing on Imitation Learning (IL) and Inverse Reinforcement Learning (IRL), the conceptual foundations of RILe.

**Imitation Learning** Early work in IL introduced Behavioral Cloning (BC) (Bain & Sammut, 1995), which frames policy learning as a supervised problem where the agent's actions are directly matched to expert demonstrations. DAgger (Ross et al., 2011) refines BC by aggregating data over multiple iterations to mitigate compounding errors. GAIL (Ho & Ermon, 2016) employs adversarial training: a discriminator learns to distinguish expert trajectories from the agent's, and takes the role of the reward function for the generator (agent). BCO (Torabi et al., 2018a) extends BC, and GAIfO

(Torabi et al., 2018b) extends GAIL, both to handle state-only observation scenarios. DQfD (Hester et al., 2018) introduces a two-stage approach with pre-training, while ValueDice (Kostrikov et al., 2020) employs a distribution-matching objective. DAC (Kostrikov et al., 2019) improves sample efficiency via off-policy learning. More recently, DRAIL (Lai et al., 2024) leverages a diffusion-based discriminator to enhance learning efficiency. TDIL (Chiang et al., 2024) introduces dense discriminator-based surrogate rewards that focus on matching expert transition dynamics.

Despite these advances, IL methods face challenges in high-dimensional environments (Peng et al., 2018; Garg et al., 2021), where the use of near-binary comparisons does not provide sufficient granular guidance.

**Inverse Reinforcement Learning**  Inverse Reinforcement Learning (IRL), introduced by Ng & Russell (2000), aims to uncover the expert's intrinsic reward function from demonstrations. IRL proceeds iteratively: it first trains a policy (the learning agent's decision-making mechanism) using the current reward function, observes how well the agent's behavior aligns with the expert's, and then refines the reward function to better guide the policy toward expert-like behaviors. Repeating this process eventually yields a reward function capable of providing nuanced feedback at different stages of learning, but the iterative process renders IRL computationally expensive. Major developments include Apprenticeship Learning (Abbeel & Ng, 2004), Maximum Entropy IRL (Ziebart et al., 2008), and adversarial variants like AIRL (Fu et al., 2018). IQ-Learn (Garg et al., 2021) reformulates IRL by integrating the inverse reward learning process into Q-learning for better scalability. More recent work focuses on unstructured data (Chen et al., 2021) and cross-embodiment transfer (Zakka et al., 2022). Recently, FM-IRL (Wan et al., 2025) improves adversarial discriminators using flow matching and derives stationary reward signals from a distribution-matching objective in a teacher–student framework. Although architecturally related, RILe is orthogonal in focus: it introduces an RL-based trainer as a meta reward learner between the discriminator or distribution-matching model and the learning agent, and learns the reward policy dynamically.

Nonetheless, IRL methods struggle with computational inefficiency and limited scalability (Arora & Doshi, 2021), particularly in high-dimensional tasks where repeated iterations of policy learning and reward refinement become costly.

**Automated Reward Shaping and Curriculum.**  Our work is also related to automated reward shaping and curriculum learning. Automated reward shaping often uses bi-level optimization or game-theoretic formulations to learn shaping functions for a fixed environment reward (Faust et al., 2019; Hu et al., 2020; Mguni et al., 2023), and empirical studies have explored how different reward shapes affect adversarial imitation learning (Wang & Li, 2021). In contrast, RILe operates without any known environment reward, learning it entirely from demonstrations. In curriculum learning, teacher-student architectures have been explored to automate task selection (Matiisen et al., 2019; Narvekar & Stone, 2019; Portelas et al., 2020). ERO (Zha et al., 2019) uses a teacher RL agent to select which past experiences the student should replay. RILe shares the meta-agent concept with ERO and teacher-curriculum learning, but operates on a fundamentally different axis: rather than filtering data or selecting tasks, RILe's trainer generates the reward values themselves on-the-fly, creating a dynamic reward landscape that guides the student from random initialization to expert.

## 3 BACKGROUND

### 3.1 MARKOV DECISION PROCESS AND REINFORCEMENT LEARNING (RL)

We consider a Markov Decision Process (MDP) defined by $(S, A, R, T, K, \gamma)$, where $S$ and $A$ are state and action spaces, $R$ is the reward function, $T(s'|s, a)$ represents transition dynamics, $\gamma$ is the discount factor, and $K(s)$ is the initial state distribution, i.e., $s_0 \sim K(s)$. Following Ho & Ermon (2016), expectation with respect to the policy $\pi \in \Pi$ refers to the expectation when actions are sampled from $\pi(s)$: $\mathbb{E}_\pi[R(s, a)] \triangleq \mathbb{E}_\pi[\sum_{t=0}^{\infty} \gamma^t R(s_t, a_t)]$. We consider a setting where $R = R(s, a)$ is parameterized by $\theta$ as $R_\theta(s, a) \in \mathbb{R}$ (Finn et al., 2016). Our work considers an imitation learning problem from expert trajectories, consisting of states $s$ and actions $a$. The set of expert trajectories $\tau_E$ are sampled from an expert policy $\pi_E$, and we assume that we have access to $m$ expert trajectories.

Reinforcement Learning (RL) seeks a policy $\pi$ that maximizes expected discounted return with entropy regularization $H(\pi) = \mathbb{E}_\pi[-\log \pi(a|s)]$ (Ho & Ermon, 2016; Bloem & Bambos, 2014):
$$\text{RL}(R_\theta(s, a)) = \pi^* = \arg\max_\pi \mathbb{E}_\pi[\sum_{t=0}^{\infty} \gamma^t R_\theta(s_t, a_t)] + H(\pi).$$

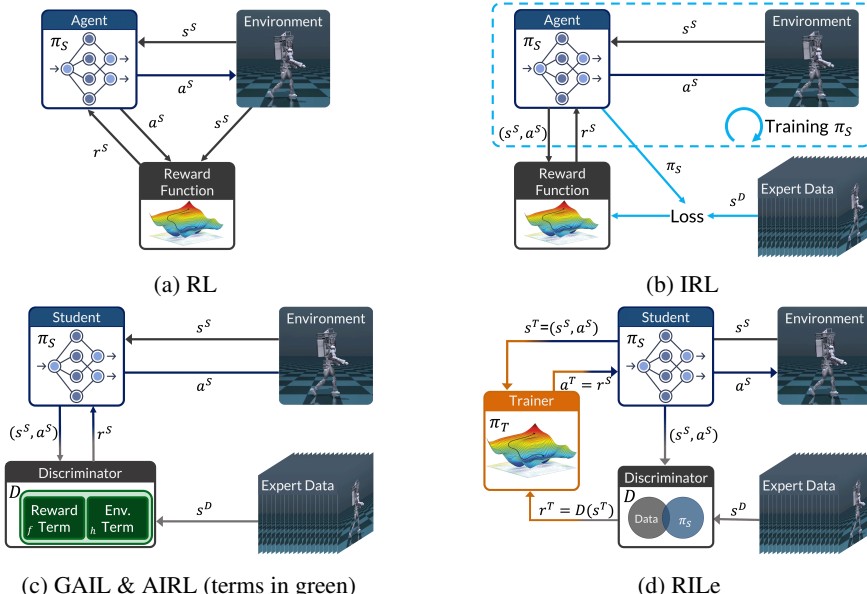

(a) RL

(b) IRL

(c) GAIL & AIRL (terms in green)

(d) RILe

Figure 1: **Overview of the related works. (a) Reinforcement Learning (RL):** learning a policy that maximizes hand-defined reward function; **(b) Inverse RL (IRL):** learning a reward function from expert data. IRL has two stages: 1. train a policy on the current reward function, and 2. update the reward function by comparing the converged policy with data. This loop is repeated multiple times; **(C) GAIL & AIRL:** using discriminator as a reward function. The discriminator is trained to distinguish agent-generated data from expert data, and the policy is trained to fool the discriminator. AIRL introduces a specific structure (terms in green) to disentangle the learned reward from environment dynamics. **(D) RILe:** learns a reward function simultaneously with the policy. The trainer agent learns to generate rewards for the student agent which learns the policy.

### 3.2 INVERSE REINFORCEMENT LEARNING (IRL)

Given trajectories $\tau_E$ from an optimal expert policy $\pi_E$, inverse reinforcement learning aims to recover a reward function $R_\theta^*(s, a)$ that maximally rewards the expert's behavior (Fig. 1-(b)). Formally, IRL seeks a reward function, $R_\theta^*(s, a)$, satisfying: $\mathbb{E}_{\pi_E}[\sum_{t=0}^\infty \gamma^t R_\theta^*(s_t, a_t)] \geq \mathbb{E}_\pi[\sum_{t=0}^\infty \gamma^t R_\theta^*(s_t, a_t) + H(\pi)] \quad \forall \pi$. Optimizing this reward function with reinforcement learning yields a policy that replicates expert behavior: $\text{RL}(R_\theta^*(s, a)) = \pi^*$. Since only the trajectories are observed, expectations over $\pi_E$ are estimated from samples in $\tau_E$. With entropy regularization $H(\pi)$, maximum causal entropy inverse reinforcement learning (Ziebart et al., 2008) is defined as

$$\text{IRL}(\tau_E) = \underset{R_\theta(s,a)\in\mathbb{R}}{\arg\max} \left( \mathbb{E}_{s,a\in\tau_E}[R_\theta(s,a)] - \max_\pi \left( \mathbb{E}_\pi[R_\theta(s,a)] + H(\pi) \right) \right). \tag{1}$$

### 3.3 ADVERSARIAL IMITATION LEARNING (AIL) AND ADVERSARIAL INVERSE REINFORCEMENT LEARNING (AIRL)

Imitation Learning (IL) directly approximates the expert policy from given expert trajectory samples $\tau_E$. It can be formulated as $\text{IL}(\tau_E) = \arg\min_\pi \mathbb{E}_{(s,a)\sim\tau_E}[L(\pi(\cdot|s), a)]$, where $L$ is a loss function, that captures the difference between policy and expert data.

GAIL (Ho & Ermon, 2016) introduces an adversarial imitation learning setting by quantifying the difference between the agent and the expert with a discriminator $D_\phi(s, a)$, parameterized by $\phi$ (Fig. 1-(c)). The discriminator distinguishes between between expert-generated state-action pairs $(s, a) \in \tau_E$ and non-expert ones $(s, a) \notin \tau_E$. The goal of GAIL is to find the optimal policy that fools the discriminator. The optimization is formulated as a zero-sum game between the discriminator $D_\phi(s, a)$ and the policy $\pi$:

$$\min_\pi \max_\phi \mathbb{E}_\pi[\log D_\phi(s, a)] + \mathbb{E}_{\tau_E}[\log(1 - D_\phi(s, a))] - \lambda H(\pi). \tag{2}$$

Consequently, the reward function that is maximized by the policy is defined as $R(s,a) = -\log\left(D_\phi(s,a)\right)$.

AIRL (Fu et al., 2018) extends AIL aiming to recover a reward function decoupled from environment dynamics (Fig. 1-(c)). AIRL structures the discriminator as:

$$D_{\phi,\psi}(s,a,s') = \frac{\exp(f_\phi(s,a,s'))}{\exp(f_\phi(s,a,s')) + \pi(a|s)}, \tag{3}$$

where $f_\phi(s,a,s') = r_\psi(s,a) + \gamma V_\phi(s') - V_\phi(s)$. Here, $r_\psi(s,a)$ represents the learned reward function that is decoupled from the environment dynamics, and $\gamma V_\phi(s') - V_\phi(s)$ is the discriminator based shaping term. The AIRL optimization problem is formulated equivalently to GAIL (see Eqn. 2). Therefore, the reward function $r_\psi(s,a)$ is learned through minimizing the cross-entropy loss inherent in this adversarial setup, and its optimization is therefore constrained by the primary objective of classification.

## 4 RILe: Reinforced Imitation Learning

We propose Reinforced Imitation Learning (RILe) to jointly learn a reward function and a policy that emulates expert-like behavior within a single learning process. RILe introduces a novel trainer–student dynamic, as illustrated in Figure 2.

In RILe, the student agent learns an action policy by interacting with the environment, while the trainer agent learns a reward function that effectively guides the student toward expert-like behavior. Both agents are trained simultaneously via reinforcement learning, with assistance from an adversarial discriminator. Specifically, the trainer queries the discriminator, which judges how expert-like the student's behavior is, and then learns a reward function based on that feedback on-the-fly. Unlike traditional AIL, where the discriminator is directly employed as the reward function, RILe introduces a trainer agent to learn a more adaptive reward function that provides context-sensitive feedback, while avoiding IRL's iterative computational expense.

The trainer agent plays the key role in RILe. Trained via RL, it learns a policy for generating rewards by maximizing the cumulative rewards it receives from the discriminator. This approach equips RILe with two key advantages: (1) **On-the-fly Reward Function Learning via RL:** The reward function is learned continuously with RL, enabling the trainer to efficiently explore different reward options and learn an effective reward function, and (2) **Context-sensitive Guidance:** The trainer continuously adjusts its reward outputs in response to the student's evolving policy, providing tailored feedback at different stages of training (see Appendix D for more discussion).

In the remainder of this section, we define the components of RILe and explain how they jointly learn from expert demonstrations.

**Student Agent** The student agent learns a policy $\pi_S$ within a standard MDP framework. Instead of using a handcrafted reward, the student is guided by the trainer's policy, $\pi_T$. At each step, the trainer's policy outputs a scalar action, $a^T \sim \pi_T((s^S, a^S))$, which directly serves as the student's reward: $r^S = a^T$. The student's objective is to maximize the expected rewards generated by the trainer:

$$\max_{\pi_S} \mathbb{E}_{(s^S,a^S)\sim\pi_S}[\pi_T\left((s^S,a^S)\right)] + \alpha H(\pi_S). \tag{4}$$

**Discriminator** The discriminator, parameterized by $\phi$, differentiates between expert-generated state-action pairs, $(s,a) \sim \tau_E$, and pairs from the student, $(s,a) \sim \pi_S$. Its objective is the standard binary cross-entropy loss from AIL:

$$\max_{\phi} \mathbb{E}_{(s,a)\sim\tau_E}[\log(D_\phi(s,a))] + \mathbb{E}_{(s,a)\sim\pi_S}[\log(1 - D_\phi(s,a))]. \tag{5}$$

As established by GAIL (Ho & Ermon, 2016), this objective effectively trains the discriminator to identify expert-like behavior.

**Trainer Agent** The trainer agent learns a policy, $\pi_T$, that outputs reward signals for the student. The trainer observes the student's current state-action pair $s^T = (s^S, a^S)$ and outputs a scalar action

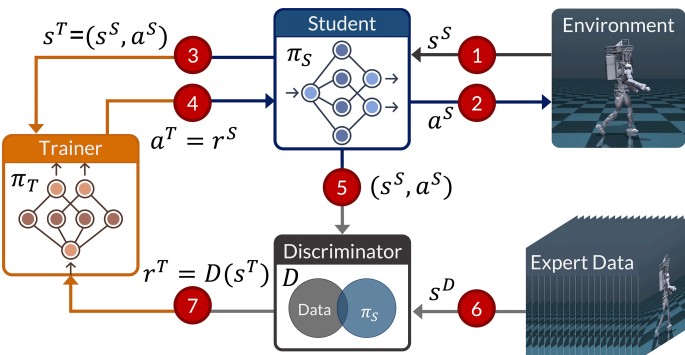

Figure 2: **Reinforced Imitation Learning (RILe)**. The framework consists of a student, trainer, and discriminator. (1-2) The student $\pi_S$ interacts with the environment. (3) The student's state-action pair $(s^S, a^S)$ becomes the trainer's observation. (4) The trainer's policy $\pi_T$ outputs an action $a^T$, which serves as the student's reward ($r^S = a^T$). (5-6) The discriminator receives both student and expert data. (7) The discriminator then provides a reward to the trainer based on the similarity between the student and expert data.

$a^T \in [-1, 1]$, which becomes the student's reward, $r^S$. The trainer is, in turn, rewarded based on how well its action $a^T$ matches the discriminator's evaluation of the student's behavior, $D_\phi(s^T)$:

$$R^T = e^{-|v(D_\phi(s^T)) - a^T|} \tag{6}$$

where $v(x) = 2x - 1$ scales the discriminator's output, making it symmetric around zero. We use exponential function since it is bounded and saturates for large deviations, which avoids over-penalizing outliers. The trainer's RL objective drives it to learn a policy that maximizes long-term outcomes, rather than myopically mimicking the discriminator at each step. Formally, the trainer's objective is to maximize its expected cumulative reward:

$$\max_{\pi_T} \mathbb{E}_{\substack{(s,a) \sim \pi_S \\ a^T \sim \pi_T}} [e^{-|v(D_\phi(s^T)) - a^T|}] + \alpha H(\pi_T). \tag{7}$$

In standard AIL, the reward function is a fixed transformation of the discriminator output, $r_g(s, a) = g(D_\phi(s, a))$, which is inherently *myopic*: it depends only on the current discriminator score and ignores how a reward at time $t$ influences future trajectories of the student. In RILe, the student reward $r_S$ is instead the action of the trainer policy $\pi_T$ that optimizes a discounted sum of discriminator-based feedback (Eq. 9). The trainer's optimal action depends not only on $D_\phi(s, a)$ but also on the environment dynamics and the current student policy, and we hypothesize that it cannot, in general, be represented as any static mapping $g(D_\phi)$ (except in the degenerate myopic case $\gamma_T=0$). Our transformed-GAIL ablations in Sec. 5.1, which directly feed these static transformations of $D_\phi$ as rewards, empirically support this, and we provide further discussion in Appendix D.

**RILe** RILe optimizes these three components, student, trainer, and discriminator, simultaneously. While both $\pi_S$ and $\pi_T$ can be trained with any single-agent RL algorithm, we use Soft Actor-Critic (SAC) (Haarnoja et al., 2018) for both policies in our implementation.

Training these components jointly introduces stability challenges inherent to multi-agent and adversarial systems. To ensure stable learning, we employ strategies such as periodically freezing the trainer's policy, which allows the student to learn from a temporarily stationary reward function. These practical techniques are important for stability and are detailed in Appendix B.

The complete optimization problem involves finding the optimal policies $\pi_S^*$ and $\pi_T^*$. The student agent aims to recover the optimal policy $\pi_S^*$:

$$\pi_S^* = \arg\max_{\pi_S} \mathbb{E}_{(s^S, a^S) \sim \pi_S} \left[ \sum_{t=0}^{\infty} \gamma^t [\pi_T((s_t^S, a_t^S)) + \alpha H(\pi_S(\cdot|s_t^S))] \right]. \tag{8}$$

Simultaneously, the trainer aims to recover $\pi_T^*$:

$$\pi_T^* = \arg\max_{\pi_T} \mathbb{E}_{\substack{s^T \sim \pi_S \\ a^T \sim \pi_T}} \left[ \sum_{t=0}^{\infty} \gamma^t [e^{-|v(D_\phi(s_t^T)) - a_t^T|} + \alpha H(\pi_T(\cdot|s_t^T))] \right]. \tag{9}$$

## 5 EXPERIMENTS

We evaluate RILe across six studies: (1) a comparison of the trainer with static reward transformations, (2) an empirical analysis of the learned reward landscapes of RILe and baselines, (3) a study on trainer design choices, (4) a comparison of computational cost against AIL and IRL, (5) a benchmark on continuous control tasks and (6) an analysis on noise robustness and covariate shift. Further experiments, including an analysis of advanced discriminators and trainer-discriminator relations, are provided in Appendix A. Experimental details and hyperparameter selections are in Appendix C and G.

**Baselines** We compare RILe with nine baseline methods: Behavioral cloning (BC (Bain & Sammut, 1995; Ross & Bagnell, 2010), BCO (Torabi et al., 2018a)), adversarial imitation learning (GAIL (Ho & Ermon, 2016), GAIfO (Torabi et al., 2018b) and DRAIL (Lai et al., 2024)), adversarial inverse reinforcement learning (AIRL (Fu et al., 2018)), and inverse reinforcement learning (GCL (Finn et al., 2016), REIRL (Boularias et al., 2011), IQ-Learn (Garg et al., 2021)).

### 5.1 TRAINER AGENT VS. STATIC TRANSFORMATIONS

We test whether simply reshaping the discriminator reward with static transformations can match RILe's performance on Humanoid-v2. We create two GAIL variants using static transformations: (1) GAIL-Exp., which uses the static reward transformation identical to the trainer's objective ($r = e^{-|1-(2D-1)|}$), and (2) GAIL-Scaled, which uses a linear scaling transformation ($r = 2D - 1$).

Figure 3 shows the results. Both static-transformation variants fail to match the performance of RILe. This suggests that RILe's improvement is not explained by simple reward shaping, but by the trainer's non-myopic guidance. We also find that replacing the discriminator with a Maximum Mean Discrepancy (MMD) evaluator (RILe-MMD) results in a functional but lower-performing policy. This indicates that the framework is compatible with alternative evaluators, but adversarial discriminators currently result in higher performance.

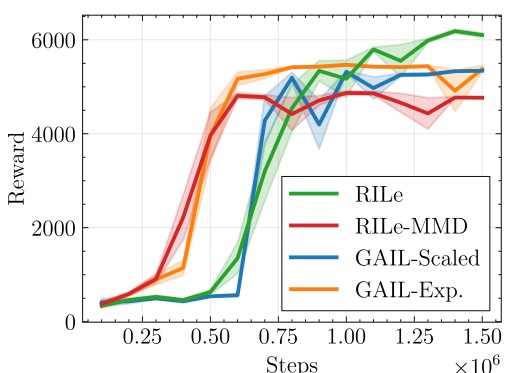

Figure 3: **Comparison of the Trainer with Static Transformations.** Learning curves on Humanoid-v2 comparing RILe against GAIL with static reward transformations (Exp., Scaled) and RILe with an MMD evaluator.

### 5.2 REWARD FUNCTION EVALUATION

To evaluate how RILe's reward-learning strategy differs from AI(R)L baselines, we compare them empirically by visualizing the learned reward functions in a maze environment. In this environment, the agent must navigate from a fixed start to a goal while avoiding static obstacles; we use a single expert demonstration.

Figure 4 shows how the learned reward function evolves during training. For RILe, we plot the reward function learned by the trainer; for GAIL and AIRL, we visualize the discriminator outputs; for TDIL, we visualize the transition-discriminator outputs. From a reward-density perspective, RILe dynamically adapts to the current student policy, providing context-sensitive guidance: early on, non-zero rewards cover a broad region around feasible paths, providing informative feedback even when the student is far from the expert, and later rewards concentrate around the expert path as the policy improves. In contrast, GAIL, AIRL, and TDIL yield comparatively static landscapes where high-reward regions remain static, offering less context-sensitive guidance to the student.

### 5.3 TRAINER DESIGN

We study how trainer design choices influence RILe's performance on MuJoCo Humanoid-v2. We vary the following hyperparameters of the trainer agent: (a) buffer size, (b) discount factor $\gamma_T$, (c)

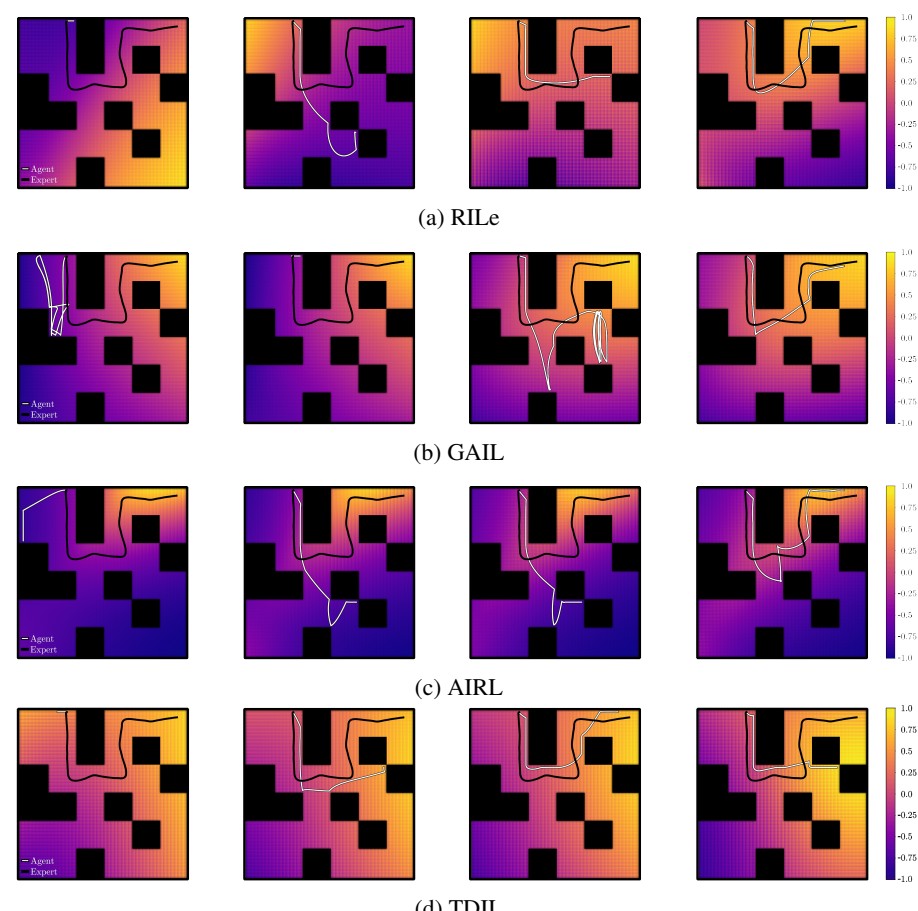

Figure 4: **Reward Function Comparison**. Evolution of reward functions during training for (a) RILe, (b) GAIL, (c) AIRL, and (d) TDIL in a maze environment. Columns show reward landscapes at 25%, 50%, 75%, and 100% of training completion. The expert's trajectory is shown in black, and the student's trajectory from the previous epoch is in white. Color gradients represent reward values.

network capacity, (d) update frequency (trainer updates per student update), (e) entropy weight $\alpha$, and (f) freezing timestep (after which only the student continues learning).

Figure 5 shows that RILe reaches high returns on a wide range of trainer configurations, although specific choices increase asymptotic performance. (a) Buffer size: a smaller buffer results in the best final returns. We hypothesize that a smaller buffer keeps the trainer better aligned with the highly dynamic student. (b) Discount factor: larger $\gamma$ makes the trainer more far-sighted and results in a higher asymptotic performance, while smaller $\gamma$ converges quickly but to lower rewards closer to GAIL. (c) Network capacity: larger networks (e.g. [256,256]) achieve the best final performance. (d) Update frequency: modestly increasing the update frequency improves performance, but too aggressive updating slightly harms final rewards, likely because the trainer converges earlier. (e) Entropy weight: a higher $\alpha$ results in fast early learning but slightly lower asymptotic performance, while a moderate entropy achieves the best final returns. (f) *Freeze time:* delaying the freezing until convergence ($\sim$ 1000k steps) helps the trainer adapt more and results in higher performance.

## 5.4 COMPUTATIONAL COST AND PERFORMANCE TRADE-OFFS

We compare RILe's computational cost with Adversarial Imitation Learning (AIL) (GAIL (Ho & Ermon, 2016), DAC (Kostrikov et al., 2018)), Inverse Reinforcement Learning (IRL) (GCL (Finn et al., 2016) and REIRL (Boularias et al., 2011)), and Distribution Matching (ValueDICE (Kostrikov et al., 2020)). The evaluation is performed on four continuous control tasks from MuJoCo Playground (Zakka et al., 2025). As gradient steps aggregated over all components are a hardware-agnostic proxy for wall-clock time, we use this metric for a fair comparison of computational cost.

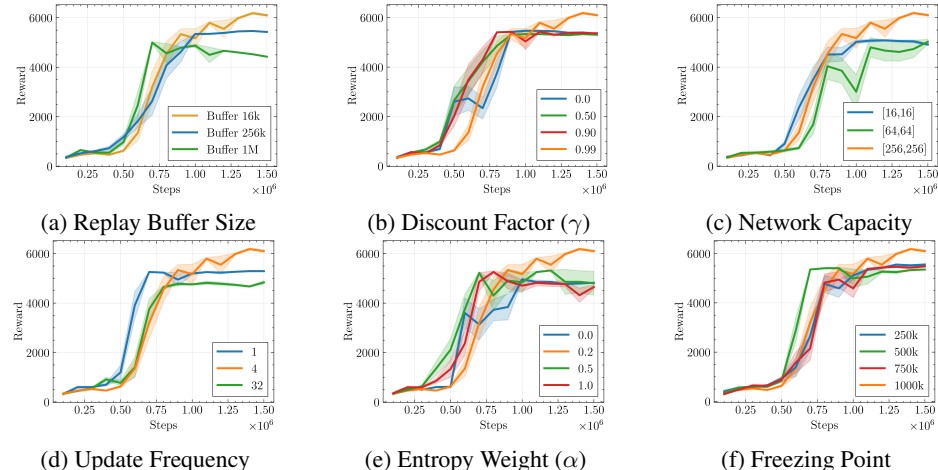

(a) Replay Buffer Size  (b) Discount Factor ($\gamma$)  (c) Network Capacity

(d) Update Frequency  (e) Entropy Weight ($\alpha$)  (f) Freezing Point

Figure 5: **Trainer Hyperparameter Analysis:** Each subplot shows RILe's learning curves when varying a single trainer hyperparameter while keeping all others fixed.

Figure 6 plots evaluation rewards against gradient steps. While GAIL is highly efficient, its peak performance is limited, particularly in the complex HumanoidStand task. Conversely, IRL methods achieve higher rewards but require orders of magnitude more gradient steps to do so. RILe successfully bridges this gap, achieving the high performance characteristic of IRL while maintaining sample efficiency closer to that of AIL.

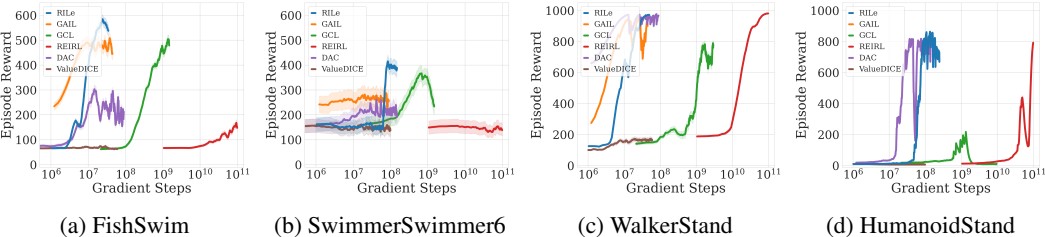

(a) FishSwim  (b) SwimmerSwimmer6  (c) WalkerStand  (d) HumanoidStand

Figure 6: **Comparison of Computational Cost:** Computational cost versus performance for RILe, GAIL, DAC, GCL, REIRL, and ValueDICE across four MuJoCo Playground tasks.

## 5.5 PERFORMANCE IN CONTINUOUS CONTROL TASKS

We evaluate RILe's performance on two sets of continuous control benchmarks. First, we use a standard benchmark of MuJoCo tasks (Todorov et al., 2012; Brockman et al., 2016), where the agent learns from a small set of clean state-action expert trajectories. As shown in Table 1a, RILe achieves the highest final reward in three of the four environments, while IQ-Learn performs best on Walker2d. IQ-Learn's direct, value-based formulation may offer stability advantages on lower-dimensional tasks like Walker2d, whereas RILe's adaptive reward mechanism appears to provide a greater benefit in high-dimensional settings like Humanoid.

Table 1: Performance in Continuous Control Tasks. Scores are averaged over test seeds. (See Table 6 and 7 for standard errors.)

(a) MuJoCo Benchmark

| | RILe | GAIL | AIRL | IQ |
|---|---|---|---|---|
| Humanoid | **5928** | 5709 | 5623 | 5258 |
| Walker2d | 4506 | 4906 | 4823 | **5133** |
| Hopper | **3573** | 3361 | 3014 | 3504 |
| HalfCheetah | **5205** | 4173 | 3991 | 4820 |

(b) LocoMujoco Benchmark

| | | RILe | GAIL | AIRL | IQ | BCO | GAIfO |
|---|---|---|---|---|---|---|---|
| Walk | Atlas | **870.6** | 792.7 | 300.5 | 30.9 | 21.0 | 834.2 |
| | Talos | **842.5** | 442.3 | 102.1 | 4.5 | 11.9 | 710.0 |
| | UnitreeH1 | **966.2** | 950.2 | 568.1 | 8.8 | 34.8 | 526.8 |
| | Humanoid | **831.3** | 181.4 | 80.1 | 4.5 | 3.5 | 706.5 |
| Carry | Atlas | **850.8** | 669.3 | 256.4 | 36.8 | 20.3 | 810.1 |
| | Talos | **220.1** | 186.3 | 134.2 | 10.5 | 10.3 | 212.5 |
| | UnitreeH1 | **788.3** | 634.6 | 130.5 | 14.4 | 21.1 | 604.5 |

Second, we test RILe on a more challenging high-dimensional robotic locomotion benchmark that uses noisy, state-only motion-capture data (Al-Hafez et al., 2023). Table 1b shows that RILe obtains the highest score across all seven tasks, which span different robotic embodiments and objectives.

The performance across both benchmarks highlights RILe's effectiveness in learning from expert data in high-dimensional domains, with both clean state-action data and more challenging noisy, state-only observations data.

## 5.6 Robustness to Noise and Covariate Shift

We evaluate RILe's robustness to noisy expert demonstrations and environmental covariate shift in Humanoid-v2. First, we inject zero-mean Gaussian noise with standard deviation $\Sigma$ into either expert actions or states. We compare RILe against GAIL (Ho & Ermon, 2016), AIRL (Fu et al., 2018), and robust baselines including TDIL (Chiang et al., 2024), RIL-Co (Tangkaratt et al., 2021), and IC-GAIL (Wu et al., 2019). Table 2 shows the results. RILe maintains high performance across all noise levels. TDIL is especially robust to action noise, since its transition discriminator depends only on state transitions and not on expert actions, but its overall performance remains below RILe.

Table 2: Robustness to different noise levels in the expert demonstrations.

|  | | RILe | GAIL | AIRL | TDIL | RIL-Co | IC-GAIL |
|---|---|---|---|---|---|---|---|
| No Noise | $\Sigma = 0$ | **5928** | 5709 | 5623 | 5268 | 576 | 610 |
| Action | $\Sigma = 0.2$ | **5280** | **5275** | 4869 | **5268** | 491 | 601 |
|  | $\Sigma = 0.5$ | 5154 | 902 | 4589 | **5268** | 493 | 568 |
| State | $\Sigma = 0.2$ | **5350** | 5147 | 4898 | 5157 | 505 | 590 |
|  | $\Sigma = 0.5$ | **5205** | 917 | 4780 | 5113 | 501 | 591 |

Table 3: Robustness to covariate shifts.

|  | RILe | AIRL |
|---|---|---|
| No Noise | **5928** | 5623 |
| Mild | | |
| $\Sigma = 0.2$ | **5201** | 5005 |
| High | | |
| $\Sigma = 0.5$ | **5196** | 4967 |

Second, inspired by Xu et al. (2022), we evaluate the stability of the reward functions learned by RILe and AIRL. First, we train both models in a noise-free environment. Then, we freeze the learned reward functions and train new student agents in environments where Gaussian noise is injected into their actions (covariate shift). Table 3 shows that the reward function learned by RILe has a better robustness to covariate shift, maintaining high performance under increased noise levels.

## 6 Discussion

Our results show that RILe is a compelling and effective alternative to established imitation learning methods. By replacing static reward signals with an adaptive trainer, RILe provides dynamic, context-sensitive guidance, which leads to strong performance in high-dimensional control tasks and is visible in the maze environment, where the reward landscape adapts to the student's progress. This dynamic guidance is particularly important in settings where a static signal can be uninformative.

The dynamic, two-level learning framework of RILe introduces a trade-off between adaptivity and stability. While practical solutions like freezing the trainer agent offer stability, they prevent continuous co-adaptation with the student. RILe also inherits challenges from its adversarial component, such as potential discriminator overfitting. Our RILe-MMD ablation suggests that the framework is compatible with non-adversarial, distance-based evaluators but naive replacements underperform discriminators in complex tasks. A promising direction for future work is to design better distance-based or metric critics for RILe (e.g., MMD, Wasserstein, or other distribution-matching objectives), and to explore trainers that leverage richer signals such as value estimates or uncertainty, rather than a single scalar reward. Finally, we view a full characterization of the three-agent equilibrium and convergence properties as an exciting direction for future work.

In conclusion, RILe frames imitation learning as a simultaneous process of learning a policy and learning how to teach it. This approach is more computationally efficient than traditional IRL and provides more adaptive feedback than AIL. By reframing the reward generator from a static judge to an adaptive coach, our work opens new possibilities for creating more intelligent and responsive learning agents.

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

# A ADDITIONAL EXPERIMENTS

## A.1 TRAINER-DISCRIMINATOR RELATION

We investigate how the interaction between the trainer agent and the discriminator affects RILe's performance by comparing different trainer reward functions. Each reward function defines a different relationship between the trainer's action $a^T$ and the discriminator's output $D_\phi(s^T)$. We consider following reward functions: (a) Difference ($R^T = -|v(D_\phi(s^T)) - a^T|$), (b) Exponential Difference (default in RILe): $R^T = e^{-|v(D_\phi(s^T)) - a^T|}$, (c) Multiplication ($R^T = v(D_\phi(s^T))a^T$), (d) Naive ($R^T = D_\phi(s^T)$), (e) Exponential Naive ($R^T = e^{1-D_\phi(s^T)}$) and (f) Sigmoid ($R^T = D_\phi(s^T)\sigma(a^T)$), where $v(x) = 2x - 1$ and $\sigma(x) = \frac{1}{1+e^{-x}}$.

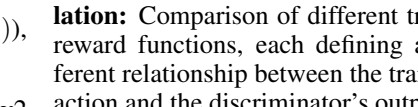

Figure 7: **Trainer-Discriminator Relation:** Comparison of different trainer reward functions, each defining a different relationship between the trainer's action and the discriminator's output.

Figure 7 presents reward curves in MuJoCo Humanoid-v2 environment. While the exponential naive reward offers the fastest convergence, the exponential difference reward offers the best performance. Therefore, we use exponential difference reward as the default reward function in RILe.

## A.2 COMPARISON WITH MULTI-STUDENT GAIL

To test whether RILe's gains can be attributed to training multiple GAIL policies and selecting the best one, we construct *multi-GAIL* baselines on Humanoid-v2. In GAIL-2 and GAIL-5, we train 2 and 5 independent GAIL agents, respectively, each with a different random seed and separate replay buffer, with a common discriminator. After training, we report the best-performing policy across the ensemble.

As Table 4 shows, despite using multiple independently trained policies and additional compute, GAIL-2 and GAIL-5 do not match RILe's performance, and in fact underperform the standard single-student GAIL baseline. We hypothesize that this performance drop stems from the shared discriminator, which becomes more robust due to the higher variety of generated data. In addition, students receive rewards shaped for the mixture rather than their own policy. This suggests that RILe's advantage is not because of an increased learning capacity, but stems from the trainer's ability to adaptively shape rewards for the student.

Table 4: Comparison against Naive GAIL Ensembles in Humanoid-v2.

| Method | Architecture | Score |
|---|---|---|
| **RILe (Ours)** | Trainer + Student | **5928** |
| GAIL-2 | Naive Ensemble (2 Agents) | 5447 |
| GAIL-5 | Naive Ensemble (5 Agents) | 5470 |

## A.3 IMPACT OF ADVANCED DISCRIMINATORS

The quality of the discriminator is critical for adversarial imitation learning. To analyze how different discriminators affect methods, we replace the standard discriminators of RILe and GAIL with the more advanced diffusion-based model from DRAIL Lai et al. (2024).

We analyze the impact on reward dynamics using three metrics, which are formally defined in Appendix C: (1) Reward Function Distribution Change (RFDC) to measure the overall volatility of the reward function, (2) Fixed-State RFDC (FS-RFDC) to measure reward volatility on a fixed set of expert data, and (3) Correlation between Performance and Reward (CPR) to measure how well reward improvements align with gains in student performance.

As shown in Figure 8, the DRAIL discriminator reduces the volatility of the reward function for both methods (lower RFDC and FS-RFDC), but RILe's reward function remains more adaptive. Furthermore, both RILe and DRAIL-RILe exhibits a positive CPR, indicating its adaptive rewards are well-aligned with student performance gains. In contrast, GAIL-based methods eventually develop a negative correlation, suggesting their more static reward signals become misaligned with the student's progress as it improves .

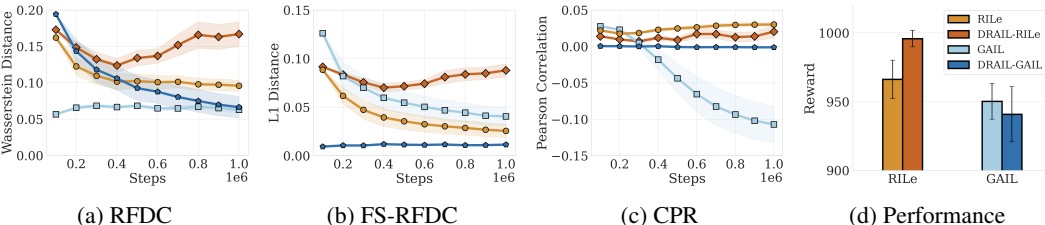

|  (a) RFDC  |  (b) FS-RFDC  |  (c) CPR  |  (d) Performance  |

Figure 8: **Impact of an Advanced Discriminator** Plots (a-c) show reward dynamics. Plot (d) compares final performance. RILe effectively leverages the advanced discriminator to improve performance.

The difference in reward dynamics appears impact final task performance (Figure 8d). With the advanced discriminator, RILe's performance improves, whereas GAIL's performance remains stable.

## B  TRAINING STRATEGIES

The introduction of the trainer agent into the AIL framework introduces instabilities that can hinder the learning process. To address these challenges, we employ three strategies.

**Freezing the Trainer Agent Midway:** We hypothesize that continuing to train the trainer agent throughout the entire process leads to overfitting on minor fluctuations in the student's behavior. This overfitting causes the trainer to assign inappropriate negative rewards, which diverts the student away from expert behavior, especially since the student agent may fail to interpret these subtle nuances correctly in the later stages of training. To prevent this, we freeze the trainer agent (and the discriminator) once its critic network within the actor-critic framework converges during the training process.

**Utilizing a Smaller Buffer for the Trainer Agent:** We employ distinct replay buffer sizes for the student and trainer agents. We use larger buffer for the student compared to the trainer, as detailed in our hyperparameter configurations (see Appendix G). This strategy ensures the trainer primarily learns from the student's recent interactions, allowing it to adapt its reward function more rapidly to the evolving student policy instead of optimizing based on potentially outdated historical data. This increased responsiveness provides more relevant, timely feedback to the student, which we found empirically contributes to more stable and effective co-adaptation within the RILe framework across different tasks.

**Increasing the Student Agent's Exploration:** We increase the exploration rate of the student agent compared to standard AIL methods. We implement an epsilon-greedy strategy within the actor-critic framework, allowing the student to occasionally take random actions. This increased exploration enables the student to visit a wider range of state-action pairs. Consequently, the trainer agent receives diverse input, helping it learn a more effective reward function. This diversity is crucial for the trainer to observe the outcomes of various actions and to guide the student more effectively toward expert behavior.

## C  EXPERIMENTAL SETTINGS

### C.1  TRAINER-DISCRIMINATOR RELATION

We use single expert trajectory and train RILe with different trainer reward functions in Humanoid-v2 environment. We show the reward curve in averaged across three evaluation seeds different from the training seeds.

### C.2 REWARD FUNCTION EVALUATION

We use single expert demonstration in this experiment, and use PPO (Schulman et al., 2017) as the learning agent for all methods. For RILe, we plot the reward function learned by the trainer. For GAIL, we visualize the discriminator output, and for AIRL, the reward term under the discriminator.

The 2D axes in the maze environment (Fig. 4) represent the state space $s^T = (s^S, a^S)$ of the trainer agent. To visualize the reward $R(s^T)$, for each student state $s^S$, we calculate reward for a fixed set of actions: $\mathcal{A} = \{(v_x, v_y) : v_x, v_y \in \{-1, -0.5, 0, 0.5, 1\}\}$. For each $s^S$ (for each x,y coordinate in the maze), we compute reward outputs for 25 actions, resulting in 5×5 slice of the reward surface. This provides a landscape of the rewards the agent could expect to receive across the maze. Every episode is initialized at the same starting point, and the training completion percentage refers to the fraction of total training steps completed.

### C.3 COMPUTATIONAL COST AND PERFORMANCE TRADE-OFFS

We train each method in MuJoCo playground, using 100 expert trajectories. Experts are trained using SAC, following Brax configurations provided in (Zakka et al., 2025; Freeman et al., 2021). We use the same configurations while training RILe, GAIL, GCL and REIRL, and train each method for 50 million steps. We plot reward curves averaged across 10 evaluation seeds, which are different from the training seeds.

### C.4 PERFORMANCE IN CONTINUOUS CONTROL TASKS

For MuJoCo, each method is trained using 25 expert trajectories provided in the IQ-Learn paper Garg et al. (2021). We use SAC as the learning agent for all methods for a fair comparison. We evaluate methods with 3 different seeds and report the mean of the results.

For LocoMujoco, we use all the motion capture data provided in (Al-Hafez et al., 2023) for training all the methods. We use SAC as the learning agent for all methods for a fair comparison. We evaluate methods across 10 different seeds, and report the average score achieved. We use different random seeds from those in training, introducing new random variations that affect the environment's dynamics during state transitions.

### C.5 IMPACT OF ADVANCED DISCRIMINATORS

In this experiment, we select the student agent's hyperparameters to be identical to those used in GAIL, ensuring that the only difference between the agents is the reward function. Therefore, we use the best hyperparameters identified for GAIL, applied to both GAIL and RILe, from our hyperparameter sweeps presented in Appendix G. We evaluate methods with 10 different seeds and report the mean of the results.

**RFDC:** We calculate the Wasserstein distance between reward distributions over consecutive 10,000-step training intervals, denoted as times $t$ and $t + 10,000$. This metric quantifies how much the overall reward distribution shifts over time. Changes in reward distributions depend both on the reward function and the student policy updates. Since we use the same student agent with the same hyperparameters, higher RFDC values still indicate that the reward function is adapting more dynamically in response to the student's learning progress.

**FS-RFDC:** We compute the mean absolute deviation of rewards between consecutive 10,000-step training intervals for a fixed set of states derived from expert data. As the fixed set, we use all the states in the expert data. Since the states used for calculating rewards are fixed, changes in this value purely depend on the reward function updates. This metric assesses how the reward values for specific states change over time.

**CPR:** We evaluate how changes in the reward function correlate with improvements in student performance. We store rewards from both the learned reward function and the environment-defined rewards in separate buffers. In other words, we collect samples from two reward functions: the learned reward function and the environment-defined reward function. The environment rewards consider the agent's velocity and stability. Every 10,000 steps, we calculate the Pearson correlation

between these rewards and empty the buffers. This metric evaluates whether increases in the learned rewards relate to performance enhancements.

# D  DISCUSSION ON RILE

## D.1  MOTIVATION FOR THE TRAINER AGENT

The fundamental limitation of adversarial learning approaches lies in the nature of their objective functions. The reward signal in AIL is a direct byproduct of a *myopic*, binary classification objective aimed at instantaneously separating expert and student data. The optimal discriminator converges to a quasistatic function of the expert and policy densities, $D^*(s, a) = p_E(s, a)/(p_E(s, a) + p_{\pi_S}(s, a))$ (Goodfellow et al., 2014). A reward derived statically from this function is also *myopic*, tends to saturate once the discriminator becomes confident, providing coarse, binary-like feedback that is often insufficient for guiding an agent through complex, high-dimensional tasks. In contrast, RILe's trainer is a fully separate reinforcement learning agent whose objective is to maximize a long-horizon, discounted sum of future discriminator rewards (Eq. 9). The trainer learns a reward-generating *policy*, not just a static function of the discriminator. This allows it to provide a seemingly suboptimal reward at the current step if its value function, $Q_T$, predicts this will lead to higher discriminator scores in the long run.

The RL-based architecture also allows the trainer to *explore reward strategies*. Because the trainer $\pi_T$ is an RL agent, its reward-giving action $a^T$ is not tied to the discriminator's instantaneous judgment. It is incentivized to explore different actions (i.e., different reward values for the student) for the same state, a process encouraged by an entropy regularization term, $H(\pi_T)$, in its objective. This allows the trainer to gradually learn to steer the student into states that yield higher long-term rewards, even if the discriminator's immediate reward is low. The result is a dynamically changing reward landscape that emphasize different subgoals as the student improves, a curriculum effect that a static transformation of $D_\phi$ fails to replicate.

RILe establishes a two-level learning dynamic rather than a fully adversarial setting. The student works to maximize the rewards provided by the trainer, while the trainer learns to provide rewards that effectively guide the student toward expert-like behavior. Their goals are aligned: for the student to successfully imitate the expert and fool the discriminator. This student-trainer pair then operates within a broader adversarial game, leveraging the feedback from the discriminator which remains in a competitive relationship with the student's generated trajectories.

## D.2  JUSTIFICATION OF THE TRAINER AGENT

The reward policy $\pi_T$ learned by the RILe trainer agent is fundamentally distinct from any static transformation $g(D_\phi)$ of the discriminator's output, except in the degenerate case where the trainer's learning objective is myopic (i.e., its discount factor $\gamma_T = 0$).

**Myopic Reward:** We consider a reward $r(s_t, a_t)$ is *myopic* if it depends only on the current discriminator output $D_\phi(s_t, a_t)$ and *not* on any future transitions or on the policy's evolution.

In frameworks like GAIL or a hypothetical variant, the reward given to the student is a fixed, or *static*, transformation, $g(D_\phi)$, of the discriminator's output:

$$r_g(s_t, a_t) = g(D_\phi(s_t, a_t)) \tag{10}$$

By definition, this reward signal is myopic. The key characteristic is that its value depends only on the instantaneous output of the discriminator and is independent of future consequences and environment dynamics.

In contrast, RILe's student reward is the action of the trainer agent:

$$r_S(s_t, a_t) = a_t^T \tag{11}$$

where $a_t^T \sim \pi_T(\cdot|s_t^T)$. The trainer is a full reinforcement learning agent, and the trainer's policy $\pi_T$ is optimized to maximize its own long-horizon objective:

$$\pi_T^* = \arg\max_{\pi_T} \mathbb{E}_{\substack{s_t^T \sim \pi_S \\ a_t^T \sim \pi_T}} \left[ \sum_{t=0}^{\infty} \gamma_T^t \Big( R_t^T + \alpha\, H\big(\pi_T(\cdot \mid s_t^T)\big) \Big) \right]. \tag{12}$$

where the crucial element is the discount factor $\gamma_T > 0$.

The core of difference lies in the definition of the trainer's action-value function, $Q_T^*(s^T, a^T)$, which the policy $\pi_T^*$ maximizes. According to the Bellman equation, $Q_T^*$ is defined recursively:

$$Q_T^*(s_t^T, a_t^T) = R_t^T + \gamma_T \mathbb{E}_{s_{t+1}^T \sim P(\cdot|s_t^T, a_t^T)}\left[V_T^*(s_{t+1}^T)\right]. \tag{13}$$

The key distinction lies in the second term:

$$\gamma_T \mathbb{E}[V_T^*(s_{t+1}^T)] = \gamma_T \mathbb{E}_{s_{t+1}^S \sim T(\cdot|s_t^S, a_t^S), a_{t+1}^S \sim \pi_S(\cdot|s_{t+1}^S)}[V_T^*(s_{t+1}^T)] \tag{14}$$

This term represents the discounted value of all future states and inextricably links the trainer's current action to its long-term consequences. The distribution of the next trainer state, $s_{t+1}^T$, is a function of the environment's dynamics, $T$, and the student's current policy, $\pi_S$. Consequently, the optimal trainer action $a_t^{*T} = \arg\max_{a_t^T} Q_T^*(s_t^T, a_t^T)$ has far richer dependencies than a static function:

$$a_t^{*T} = f(D_\phi(s_t, a_t), \gamma_T, T, \pi_S)$$

Because the trainer's reward signal is dependent on its discount factor $\gamma_T$, environment dynamics $T$, and the student's policy $\pi_S$, it cannot be reduced to a static transformation $g(D_\phi)$, which lacks these dependencies. The trainer learns a strategic, forward-looking teaching policy rather than executing a reactive, myopic mapping.

The only scenario where this distinction vanishes is the degenerate case where $\gamma_T = 0$. If the trainer is myopic, the future-looking term in Equation 13 disappears. The objective collapses to maximizing the immediate reward $R_t^T$, making the trainer's action a deterministic function of $D_\phi$ and thus functionally equivalent to a static transformation.

Finally, the trainer's objective is also optimized with an entropy regularization term, $\alpha H(\pi_T)$, which forces the policy to be *stochastic*. A policy that outputs a distribution over rewards cannot be equivalent to a *deterministic* function like $g(D_\phi)$, providing a second, independent reason for their non-equivalence.

Let $g : [0, 1] \to [-1, 1]$ be any deterministic function. There exists an MDP, student policy $\pi_S$, and corresponding discriminator $D_\phi$ for which the optimal trainer action $a_T^*(s, a)$ differs between two contexts despite $D_\phi(s, a)$ being identical. Hence no static reward ($r_g(s, a) = g(D_\phi(s, a))$) can match the long-horizon shaping of $\pi_T^*$.

We construct a simple 1-step MDP. Let the state space be $\mathcal{S} = \{s_0, s_1, s_2\}$, where $s_0$ is the initial state and $s_1, s_2$ are terminal states. From $s_0$, the student takes action $a_0$. The expert demonstration is the trajectory $\tau_E = (s_0, a_0, s_2)$, establishing $s_2$ as the desirable outcome. We assume the discriminator's output for the initial state-action pair as $D_\phi(s_0, a_0) = d_0$. Within the RILe framework, trajectories ending in the expert state $s_2$ yield higher long-term cumulative rewards for the trainer than those ending in $s_1$. We thus define the trainer's terminal state values as $V_T(s_1) = V_{low}$ and $V_T(s_2) = V_{high}$, where $V_{high} > V_{low}$.

The trainer's action $a_T$ at $(s_0, a_0)$ becomes the student's reward, which influences the student's policy and thus the state transition probabilities. We model the probability of reaching the desirable state $s_2$ as a function of $a_T$ using the sigmoid function $\sigma(x) = (1 + e^{-x})^{-1}$, such that $P(s' = s_2|s_0, a_T) = \sigma(\beta a_T)$. The parameter $\beta$ models the student's responsiveness. We analyze two contexts representing different student learning stages. In Context C (Eager Student), the student responds positively to reward, so we set $\beta = k$ for some $k > 0$. In Context C' (Naive Student), the student responds perversely to reward because of heavy exploration, so we set $\beta = -k$.

The trainer chooses $a_T$ at $s_0$ to maximize its Q-value, which is the sum of the immediate reward and the discounted expected future value:

$$Q_T(s_0, a_T) = e^{-|v(d_0) - a_T|} + \gamma_T \left[P(s_2|a_T)V_{high} + (1 - P(s_2|a_T))V_{low}\right]$$

where $v(d_0) = 2d_0 - 1$. In Context C, $P_C(s_2|a_T) = \sigma(ka_T)$. To maximize $Q_T$, the trainer is incentivized to choose a high $a_T$, as this maximizes both the immediate and expected future reward terms. In Context C', $P_{C'}(s_2|a_T) = \sigma(-ka_T)$. Here, the trainer faces a trade-off: a high $a_T$ maximizes the immediate reward but minimizes the future reward. To maximize the total Q-value, the trainer must choose a low $a_T$ to steer the student to $s_2$. Since the optimal action $a_T^*$ differs between contexts for the same input $d_0$, no static function $g(d_0)$ can replicate this behavior.

# E  EXTENDED LOCOMUJOCO RESULTS

We present LocoMujoco results for the validation setting and test setting, with standard errors, in Table 5 and 6, respectively.

Table 5: Validation results on seven LocoMujoco tasks.

| | | RILe | GAIL | AIRL | IQ | BCO | GAIfO | Expert |
|---|---|---|---|---|---|---|---|---|
| Walk | Atlas | 895.4 ±25 | 918.6 ±133 | 356.0 ±68 | 32.1 ±4 | 28.7 ±4 | 831.6 ±41 | 1000 |
| | Talos | 884.7 ±8 | 675.5 ±105 | 103.4 ±22 | 7.2 ±2 | 19.9 ±4 | 718.8 ±16 | 1000 |
| | UnitreeH1 | 980.7 ±15 | 965.1 ±20 | 716.2 ±124 | 12.5 ±6 | 43.7 ±8.4 | 586.6 ±102 | 1000 |
| | Humanoid | 970.3 ±101 | 216.2 ±18 | 78.2 ±6 | 6.8 ±1 | 8.3 ±1 | 345.7 ±34 | 1000 |
| Carry | Atlas | 889.7 ±44 | 974.2 ±80 | 271.9 ±30 | 39.5 ±8 | 42.7 ±9 | 306.2 ±9 | 1000 |
| | Talos | 503.3 ±72 | 338.5 ±48 | 74.1 ±8 | 11.7 ±3 | 8.1 ±1 | 444.5 ±96 | 1000 |
| | UnitreeH1 | 850.6 ±80 | 637.4 ±90 | 140.9 ±21 | 12.3 ±2 | 30.2 ±5 | 503.6 ±55 | 1000 |

Table 6: Test results on seven LocoMujoco tasks.

| | | RILe | GAIL | AIRL | IQ | BCO | GAIfO | Expert |
|---|---|---|---|---|---|---|---|---|
| Walk | Atlas | 870.6 ±13 | 792.7 ±105 | 300.5 ±74 | 30.9 ±10 | 21.0 ±3 | 803.1 ±68 | 1000 |
| | Talos | 842.5 ±24 | 442.3 ±76 | 102.1 ±17 | 4.5 ±3 | 11.9 ±1 | 687.2 ±44 | 1000 |
| | UnitreeH1 | 966.2 ±14 | 950.2 ±13 | 568.1 ±156 | 8.8 ±3 | 34.8 ±10 | 526.8 ±72 | 1000 |
| | Humanoid | 831.3 ±98 | 181.4 ±24 | 80.1 ±9 | 4.5 ±2 | 3.5 ±2 | 292.1 ±25 | 1000 |
| Carry | Atlas | 850.8 ±62 | 669.3 ±55 | 256.4 ±47 | 36.8 ±14 | 20.3 ±1 | 402.9 ±39 | 1000 |
| | Talos | 220.1 ±88 | 186.3 ±28 | 134.2 ±18 | 10.5 ±3 | 10.3 ±2 | 212.5 ±32 | 1000 |
| | UnitreeH1 | 788.3 ±71 | 634.6 ±45 | 130.5 ±22 | 14.4 ±2 | 21.1 ±6 | 504.5 ±30 | 1000 |

# F  EXTENDED MUJOCO RESULTS

We present MuJoCo results for the test setting, with standard errors, in Table 7.

Table 7: Test results on four MuJoCo tasks with standard errors.

| | RILe | GAIL | AIRL | IQLearn |
|---|---|---|---|---|
| Humanoid-v2 | $5928 \pm 188$ | $5709 \pm 63$ | $5623 \pm 252$ | $5258 \pm 161$ |
| Walker2d-v2 | $4506 \pm 253$ | $4906 \pm 159$ | $4823 \pm 221$ | $5133 \pm 33$ |
| Hopper-v2 | $3573 \pm 153$ | $3361 \pm 51$ | $3014 \pm 190$ | $3504 \pm 63$ |
| HalfCheetah-v2 | $5205 \pm 31$ | $4173 \pm 94$ | $3991 \pm 126$ | $4820 \pm 123$ |

## G HYPERPARAMETERS

We present hyperparameters in Table 8. We use SAC Haarnoja et al. (2018) as the RL architecture for both the student and trainer agents for RILe by default. For DRAIL, we replaced the discriminators with the implementation provided by DRAIL and adopted their hyperparameters for the HandRotate task.

Our experiments revealed that RILe is more sensitive to the hyperparameters of the discriminator compared to other methods. Specifically, increasing the discriminator's capacity or training speed, by using a larger network architecture or increasing the number of updates per iteration, adversely affects RILe's performance. A powerful discriminator tends to overfit quickly to the expert data, resulting in high confidence when distinguishing between expert and student behaviors. This poses challenges for the trainer agent, as the discriminator's feedback becomes less informative.

## H COMPUTE RESOURCES

For the training of RILe and baselines, following computational sources are employed:

- AMD EPYC 7742 64-Core Processor
- 1 x Nvidia A100 GPU
- 32GB Memory

Table 8: Hyperparameter Sweeps and Best Hyperparameters for LocoMujoco and Humanoid Experiments

| | Hyperparameters | RILe | GAIL | AIRL | IQ-Learn |
|---|---|---|---|---|---|
| **Discriminator** | Updates per Round | **1**, 2, 8 | **1**, 2, 8 | **1**, 2, 8 | - |
| | Batch Size | **32**, 64, 128 | **32**, 64, 128 | **32**, 64, 128 | - |
| | Buffer Size | 8192, **16384**, 1e5 | 8192, **16384**, 1e5 | 8192, **16384**, 1e5 | - |
| | Network | [512FC, 512FC] [256FC, 256FC] [**64FC, 64FC**] | [512FC, 512FC] [256FC, 256FC] [**64FC, 64FC**] | [512FC, 512FC] [256FC, 256FC] [**64FC, 64FC**] | - |
| | Gradient Penalty | 0.5, **1** | 0.5, **1** | 0.5, **1** | - |
| | Learning Rate | 3e-4, 1e-4, **3e-5**, 1e-5 | 3e-4, 1e-4, **3e-5**, 1e-5 | 3e-4, 1e-4, **3e-5**, 1e-5 | - |
| **Student** | Buffer Size | 1e5, **1e6** | 1e5, **1e6** | 1e5, **1e6** | 1e5, **1e6** |
| | Batch Size | 32, **256** | 32, **256** | 32, **256** | 32, **256** |
| | Network | [**256FC, 256FC**] | [**256FC, 256FC**] | [**256FC, 256FC**] | [**256FC, 256FC**] |
| | Activation Function | **ReLU**, Tanh | **ReLU**, Tanh | **ReLU**, Tanh | **ReLU**, Tanh |
| | Discount Factor ($\gamma$) | **0.99**, 0.97, 0.95 | **0.99**, 0.97, 0.95 | **0.99**, 0.97, 0.95 | **0.99**, 0.97, 0.95 |
| | Learning Rate | **3e-4**, 1e-4, 3e-5, 1e-5 | **3e-4**, 1e-4, 3e-5, 1e-5 | **3e-4**, 1e-4, 3e-5, 1e-5 | **3e-4**, 1e-4, 3e-5, 1e-5 |
| | Tau ($\tau$) | 0.05, **0.01**, 0.005 | 0.05, **0.01**, 0.005 | 0.05, **0.01**, 0.005 | 0.05, **0.01**, 0.005 |
| | Epsilon-greedy | 0, 0.1, **0.2** | **0**, 0.1, 0.2 | **0**, 0.1, 0.2 | **0**, 0.1, 0.2 |
| | Entropy | **0.2**, 0.5, 1 | **0.2**, 0.5, 1 | **0.2**, 0.5, 1 | 0.05, 0.1, **0.2**, 0.5, 1 |
| **Trainer** | Buffer Size | 8192, **16384**, 1e5, 1e6 | - | - | - |
| | Batch Size | 32, **256** | - | - | - |
| | Network | [**256FC, 256FC**] [64FC, 64FC] | - | - | - |
| | Activation Function | **ReLU**, Tanh | - | - | - |
| | Discount Factor ($\gamma$) | **0.99**, 0.97, 0.95 | - | - | - |
| | Learning Rate | **3e-4**, 1e-4, 3e-5, 1e-5 | - | - | - |
| | Tau ($\tau$) | 0.05, 0.01, 0.005 | - | - | - |
| | Entropy | **0.2**, 0.5, 1 | - | - | - |
| | Freeze Threshold | 1, 0.5, **0.1**, 0.01, 0.001 | - | - | - |

# I    ALGORITHM

---

**Algorithm 1** RILe Training Process

---

1: Initialize student policy $\pi_S$ and trainer policy $\pi_T$ with random weights, and the discriminator $D$ with random weights.
2: Initialize an empty replay buffer $B$
3: **for** each iteration **do**
4:     Sample trajectory $\tau_S$ using current student policy $\pi_S$
5:     Store $\tau_S$ in replay buffer $B$
6:     **for** each transition $(s, a)$ in $\tau_S$ **do**
7:         Calculate student reward $R^S$ using trainer policy:

$$R^S \leftarrow \pi_T \tag{15}$$

8:         Update $\pi_S$ using policy gradient with reward $R^S$
9:     **end for**
10:     Sample a batch of transitions from $B$
11:     Train discriminator $D$ to classify student and expert transitions

$$\max_D E_{\pi_S}[\log(D(s,a))] + E_{\pi_E}[\log(1 - D(s,a))] \tag{16}$$

12:     **for** each transition $(s, a)$ in $\tau_S$ **do**
13:         Calculate trainer reward $R^T$ using discriminator:

$$R^T \leftarrow e^{-|v(D(s,a)) - a^T|} \tag{17}$$

14:         Update $\pi_T$ using policy gradient with reward $R^T$
15:     **end for**
16: **end for**

---

---

**Algorithm 2** RILe Training Process with Off-policy RL

---

1: Initialize student policy $\pi_S$, trainer policy $\pi_T$, and the discriminator $D$ with random weights.
2: Initialize an empty replay buffers $B_D$, $B_S$, $B_T$ with different sizes
3: **for** each iteration **do**
4:      Sample trajectory $\tau_S$ using current student policy $\pi_S$
5:      Store $\tau_S$ in replay buffers $B_D$, $B_S$, $B_T$
6:      Sample a batch of transitions, $b_S$ from $B_S$
7:      **for** each transition $(s, a)$ in $b_S$ **do**
8:          Calculate student reward $R^S$ using trainer policy:

$$R^S \leftarrow \pi_T \tag{18}$$

9:          Update $\pi_S$ using calculated rewards
10:      **end for**
11:      Sample a batch of transitions $b_D$ from $B_D$
12:      Train discriminator $D$ to classify student and expert transitions

$$\max_D E_{\pi_S}[\log(D(s,a))] + E_{\pi_E}[\log(1 - D(s,a))] \tag{19}$$

13:      Sample a batch of transitions, $b_T$ from $B_T$
14:      **for** each transition $(s, a)$ in $b_T$ **do**
15:          Calculate trainer reward $R^T$ using discriminator:

$$R^T \leftarrow e^{-|v(D(s,a)) - a^T|} \tag{20}$$

16:          Update $\pi_T$ using calculated rewards
17:      **end for**
18: **end for**

---

