# OpenReview forum: "RILe: Reinforced Imitation Learning"
_ICLR.cc/2026/Conference — Submitted to ICLR 2026_

### Official Review · Reviewer_NWQg · 2025-10-28

**Soundness:** 4
**Presentation:** 4
**Contribution:** 2
**Rating:** 4
**Confidence:** 3

**Summary:**

The paper introduces a novel trainer–student framework that unifies reinforcement learning, inverse reinforcement learning, and adversarial imitation learning. Instead of relying on a static discriminator-based reward, RILe employs a trainer agent that learns a reward policy through reinforcement learning, guided by a discriminator that evaluates expert-likeness. This enables the reward function to be learned on-the-fly, in parallel with the student’s policy, producing adaptive, temporally coherent rewards that evolve as the student improves. The approach achieves state-of-the-art performance across complex continuous control benchmarks like MuJoCo and LocoMujoco while maintaining IRL-level reward quality with AIL-level efficiency. RILe’s main contributions are: (1) a dynamic reward-learning mechanism via a trainer policy, (2) a unified and efficient formulation of imitation learning, and (3) empirical and theoretical evidence that adaptive reward shaping enhances robustness and sample efficiency.

**Strengths:**

The paper is original, introducing a novel trainer–student paradigm that transforms reward learning in imitation learning from a static, adversarial process into a dynamic, cooperative one. The approach creatively integrates ideas from RL, IRL, and AIL into a unified framework that learns both policy and reward jointly and efficiently. The technical quality is strong, with clear theoretical motivation, sound algorithmic formulation, and comprehensive experiments demonstrating consistent improvements in performance, robustness, and efficiency. The paper is also well-written and clear, effectively conveying complex multi-agent learning dynamics through intuitive explanations and visualizations. In terms of significance, RILe meaningfully advances the field by addressing long-standing issues of myopic reward learning and instability in imitation learning, offering a scalable, general framework applicable to high-dimensional control and potentially to broader meta-reward learning problems.

**Weaknesses:**

While the paper presents a compelling and well-executed framework, its core innovation—training the reward function via reinforcement learning—may be seen as an incremental extension rather than a fundamentally novel concept. Empirically, while results are strong, the analysis of reward dynamics remains primarily qualitative; quantitative metrics of reward adaptivity or temporal consistency would better substantiate the claimed non-myopic behavior. Finally, the computational overhead and stability characteristics of jointly training three networks (student, trainer, discriminator) are not fully explored, and an ablation on trainer capacity or update frequency could clarify the method’s efficiency–stability trade-offs.

**Questions:**

1.Quantify Non-Myopic Reward Behavior:
A central claim is that RILe’s trainer produces temporally coherent, non-myopic rewards. Could the authors provide a quantitative metric or visualization (e.g., temporal correlation of rewards with long-term returns, or trajectory-level variance analysis) to empirically demonstrate this adaptivity beyond qualitative plots?

2.Analyze Computational Cost and Stability:
Since RILe trains a student, trainer, and discriminator simultaneously, what is the computational overhead compared to GAIL/AIRL? How sensitive is training to hyperparameters like the trainer’s update frequency, learning rate, or capacity? An ablation study would strengthen claims of efficiency and scalability.

3.Theoretical Clarifications:
The paper argues that the trainer’s reward cannot be represented as a static transformation of the discriminator. Could the authors include a brief proof sketch or intuition in the main text (not just the appendix) to make this key point more accessible?

---

> ### Author Response · Authors · 2025-12-02
> **Response to Reviewer NWQg**
>
> We thank the reviewer for their positive assessment of our work’s originality, technical quality, clear theoretical motivation, and comprehensive experiments. We appreciate the reviewer’s questions regarding reward adaptivity and stability. We have updated the paper with new quantitative analyses and ablations to address these points.
>
> ---
>
> **1. Quantifying Non-Myopic Reward Behavior (Q1):**
>
> * **Static Transformation Study:** We have added a new study in **Section 5.1 (Trainer Agent vs. Static Transformations)** to empirically quantify the necessity of the trainer's non-myopic optimization. We trained a GAIL agent using the exact reward transformation maximized by our trainer ($r(s,a)=e^{-|1-(2D_{\phi}(s,a)-1)|}$). This static variant (**GAIL-Exp**) relies solely on the transformed immediate discriminator signal. As shown in **Figure 3**, RILe outperforms both GAIL-Exp and GAIL-Scaled on -*Humanoid-v2*. Since the reward function is identical in GAIL-Exp, this experiment confirms that the RL-based, non-myopic optimization performed by the trainer is the key factor enabling RILe to achieve better performance than baselines. In line with the reviewer’s intuition, this empirically supports the claim that optimizing long-term discriminator-based returns provides a more useful teaching signal than any greedy or static transformation.
>
> * **Gamma Ablation:** Furthermore, we analyze the trainer's discount factor $\gamma$ in **Section 5.3 (Figure 5b)**. We find that low discount factor values (myopic) degrade performance to GAIL levels, while a high $\gamma$ is critical for peak performance. This quantitatively shows that the trainer leverages future value estimates to guide the student.
>
> * **Quantitative Comparison of Learned Rewards:** In **Appendix A.3**, we show that the reward function learned by RILe is more adaptive compared to GAIL (higher RFDC/FS-RFDC), and increases in learned reward align with gains in true performance (positive CPR), whereas GAIL variants eventually develop negative CPR.
>
> ---
>
> **2. Computational Cost and Stability (W, Q2):**
>
> * **Stability Analysis:** We added a comprehensive hyperparameter study in **Section 5.3 (Figure 5)** analyzing six key trainer parameters, including buffer size, network capacity, update frequency, and freezing point. The results show that RILe is robust across a wide range of settings (e.g., it achieves high performance across various network sizes and update frequencies).
>
> * **Computational Cost:** We explicitly analyze computational overhead in **Section 5.4**. While training three networks is costlier than GAIL, **Figure 6** shows that RILe is far more sample-efficient than IRL methods and achieves higher asymptotic performance than GAIL. The efficiency is comparable to other actor-critic baselines like DAC, and significantly faster than iterative IRL loops.
>
> ---
>
> **3. Theoretical Clarifications (Q3):**
>
> Following the suggestion, we have revised the **Trainer Agent** description in **Section 4** to include the theoretical intuition directly in the main text. We explicitly state: “The trainer’s optimal action depends not only on $D_{\phi}(s,a)$ but also on the environment dynamics and the current student policy. We hypothesize that it cannot, in general, be represented as any static mapping $g(D_\phi)$ (except in the degenerate myopic case $\gamma=0$)”. We then link this theoretical insight to the empirical validation in **Section 5.1** and the full discussion in **Appendix D**.
>
> ---
>
> We believe these additions address the reviewer's concerns on non-myopic behavior and stability.

---

### Official Review · Reviewer_Ysyr · 2025-10-29

**Soundness:** 3
**Presentation:** 2
**Contribution:** 2
**Rating:** 4
**Confidence:** 4

**Summary:**

This paper proposes RILE, an algorithm for adversarial imitation learning along the lines of GAIL. The main novelty is the proposal of a trainer agent, in addition to the typical discriminator and student (imitation)  policy. The trainer’s role is to provide an adaptive reward for the student, to allow the student to learn more efficiently from the discriminator’s classification-based binary reward signal. RILE is evaluated on classic Mujoco and Mujoco locomotion benchmark tasks.

**Strengths:**

- RILE leads to small but broad improvement across the board in Mujoco+
- In addition to the positive core result, there are a good number of experiments exploring the properties of various components of RILE (e.g. the reward function comparison (Fig 4), the impact of the function transform for the trainer agent's reward (Sec 5.1), the comparison of RILE to diffusion-based discriminators (Sec 5.5), and the study of the impact of noise).
- RILE leads to significantly improved robustness to noise & covariate shift compared to GAIL (App A.1) - I think this is a pretty strong and interesting result, much more so than the supplemental analysis results presented in the main paper (e.g. the comparison to DRAIL). I suggest presenting this in the main paper and expanding the analysis in this direction.

**Weaknesses:**

- **The existing results in the paper do not fully convince me that the method is sound.** The main result (Fig 4) presents relatively small empirical improvements in Mujoco, that in my view, do not justify the significant increase in algorithm complexity that RILE presents. Unstable multi-agent dynamics are already a challenge that adversarial imitation learning methods must cope with without introducing a third agent. The fact that various empirical tricks (e.g. freezing the trainer, tuning the exploration of the learner) are necessary to make RILE work, only further strengthens this opinion.
    - For this paper, theoretical analysis would go a long ways towards convincing me of the soundness of the method. One questions I have is: (1) Can we describe the behaviors of the trainer, student, and discriminator at the Nash equilibrium (i.e. if all three agents have converged to their respective optimal policies)

- **Missing experiment: train GAIL/AIRL with the transformed discriminator reward directly.** The core idea is introducing a trainer agent that optimizes for a reward function consisting of a transformation of the discriminator-based reward. The paper argues that the trainer agent's behavior is fundamentally different from directly providing the discriminator reward because it optimizes the *long term* discriminator reward. While this is transparently true from the objectives alone, it's not immediately clear that optimizing the long term discriminator reward is good (or why it would be good). Thus, there should be an empirical comparison between RILE and training the closest baseline (so, either GAIL or AIRL) with the transformed discriminator reward directly.

- **Insufficient contextualization with related work.** Currently, only the related work in IL and IRL is discussed. These works form the foundation of the paper, so indeed, they are related. However, these topics are already covered in the introduction and background. The paper would be better positioned with respect to the literature by including a discussion of the following topics:
    - Reward shaping literature: for example, [https://arxiv.org/abs/2103.09159](https://arxiv.org/abs/2103.09159), [https://arxiv.org/pdf/2104.06687](https://arxiv.org/pdf/2104.06687))

    - Automated experience replay type algorithms, which also introduce another agent whose job is to improve the learning of the main agent (e.g. [https://www.ijcai.org/proceedings/2019/0589.pdf](https://www.ijcai.org/proceedings/2019/0589.pdf))

    - Automated Curriculum learning with RL (e.g. [https://arxiv.org/abs/1812.00285](https://arxiv.org/abs/1812.00285))

**Questions:**

There are some questions embedded in the text above. Here are some additional questions.

- Why do you compare against BCO and GAIfO? This is not an imitation learning from observation setting.

- Why does DRAIL-GAIL do slightly worse than GAIL (Fig 6d)? What is the number of seeds used for this comparison?

Here are also some typos / small writing issues I noticed.

- The end of paragraph 2 in the intro (introducing ‘AIRL’) is redundant with the part of the 3rd paragraph introducing AIL

    - By the way, using AIRL as an acronym for the general problem of adversarial inverse RL collides with the algorithm AIRL (Fu et al. 2018).

- Comment on Table 1: there should be a  pointer to the tables in the Appendix with the standard errors for Table 1 in the main text of the paper.

---

> ### Author Response · Authors · 2025-12-02
> **Response to Reviewer Ysyr (Part 1)**
>
> We thank the reviewer for their constructive feedback and for identifying the “significantly improved robustness” as a key strength. We have updated the paper to directly address the reviewer’s concerns, particularly by adding the static transformation experiment and moving the robustness analysis to the main text.
>
> ---
>
> **1. Comparison with Static Transformation (W2):**
>
> We added **Section 5.1 (Trainer Agent vs. Static Transformations)** to explicitly test this. We trained a GAIL agent using the exact reward transformation used by our trainer's objective: $r(s,a)=\exp(-|1-(2D_{\phi}(s,a)-1)|)$, which we call **GAIL-Exp**. We also trained another GAIL agent using a linear reward transformation ($r(s,a) = 2D(s,a)-1$), which we call **GAIL-Scaled**. As shown in **Figure 3**, RILe outperforms both GAIL-Exp and GAIL-Scaled. Since the reward function form is identical, this experiment suggests that the trainer's RL-based, non-myopic optimization is the key factor enabling RILe to achieve better performance than baselines. As the reviewer hypothesized, this experiment empirically shows that optimizing the long-term discriminator reward (non-myopic guidance) provides a signal that helps the student navigate the optimization landscape better than a greedy/static transformation.
>
> ---
>
> **2. Robustness to Noise & Covariate Shift (Strengths):**
>
> We moved the robustness analysis from the Appendix to **Section 5.6** of the main paper. We also expanded it to include TDIL as a robust baseline. The results (**Table 2**) show that RILe matches the robustness of specialized methods like TDIL under action noise while outperforming them in clean or state-noise settings.
>
> ---
>
> **3. Improvements vs. Added Complexity/Sensitivity (W1):**
>
> * **Performance:** Across 15 continuous control tasks (4 MuJoCo Playground, 4 MuJoCo, 7 LocoMujoco), RILe attains the best score in the majority of environments, particularly the highest-dimensional ones (*HumanoidStand*, *Humanoid-v2*, LocoMujoco *Humanoid/Atlas/Talos*), where static rewards are brittle. RILe also delivers substantial robustness benefits. **Section 5.6** (previously Appendix A.1) presents robustness to noisy demonstrations and covariate shift: RILe consistently maintains high returns under both action and state noise, and its learned reward function transfers better under covariate shift than AIRL.
>
> * **Stability Analysis:** We added a comprehensive hyperparameter study in **Section 5.3 (Figure 5)** analyzing six key trainer parameters, including buffer size, entropy, and freezing point. The results show that RILe is robust across a wide range of settings and achieves high performance over a broad range of trainer hyperparameters (buffer size, discount factor, network capacity, update frequency, entropy, freeze time).
>
> * **Efficiency:** We extended the efficiency analysis in **Section 5.4 (Figure 6)** to include **ValueDICE** [6] and **DAC** [7]. Results show that, despite introducing a third component, RILe achieves IRL-level asymptotic performance at a computational cost much closer to AIL.

---

> ### Author Response · Authors · 2025-12-02
> **Response to Reviewer Ysyr (Part 2)**
>
> **4. Contextualization with Related Work (W3):**
>
> We expanded the **Related Work** section to incorporate:
>
> * **Reward Shaping:** We discuss bi-level optimization, game-theoretic shaping for fixed environment rewards, and studies on reward shaping in AIL (e.g., [1, 2, 3]), distinguishing RILe by its ability to learn rewards from demonstrations without a ground truth.
> * **Automated Experience Replay & Curriculum:** We discuss ERO [4] and Automated Curriculum Learning [5]. We clarify that while these methods select data/tasks, RILe's trainer generates the reward signal itself on-the-fly.We emphasize that RILe is complementary to these lines of work: it can, in principle, be combined with replay optimization or curricula, but focuses specifically on meta-learning the reward policy in an adversarial imitation setting.
>
> ---
>
> **5. Theoretical Clarification & Nash Equilibrium (W1):**
>
> We view a full characterization of the three-agent equilibrium as an exciting direction for future work, and we now explicitly mention this in the Discussion. While a full Nash-equilibrium analysis is beyond the scope of this submission, we refined **Section 4** to clarify the core intuition. We explicitly contrast RILe with AIL. In AIL, rewards are a fixed transform $r=g(D_{\phi})$, which is inherently myopic. In RILe, the trainer optimizes a discounted RL objective. We hypothesize that the optimal trainer policy cannot be represented as a static map $g(D_{\phi})$ except in the degenerate case where $\gamma=0$. We link this intuition directly to the empirical results in **Section 5.1** (static-transform GAIL-Exp / GAIL-Scaled vs. RILe) and the gamma ablation in **Section 5.3**. In **Appendix D**, we further discuss this intuition and provide a Bellman-equation argument illustrating that the optimal trainer action depends on future state values, supporting that the trainer acts as a dynamic potential function rather than a static classifier.
>
> ---
>
> **6. Additional Questions:**
>
> * **BCO/GAIfO Comparison:** In LocoMujoco, the expert dataset is state-only motion-capture data. Therefore, we included these for a fair comparison and also to demonstrate generality against state-only baselines.
> * **DRAIL-GAIL Performance:** We hypothesize that the stronger diffusion-based discriminator in DRAIL can over-sharpen the reward landscape, which might increase mode collapse in complex tasks. In RILe, the trainer can filter and reshape this sharper signal into a more useful teaching signal, which is why DRAIL-RILe benefits from the diffusion discriminator while DRAIL-GAIL does not in a complex environment. We also clarify that all results in Fig. 6 use the same number of seeds as our main benchmarks.
> * **Typos:** We have corrected the "AIRL" acronym by including a reference to the AIRL paper which was missing. We added pointers for standard errors in Table 1 as requested.
>
> ---
>
> We believe the new static-transformation experiment, extended robustness analysis, and detailed contextualization address the reviewer's concerns.
>
> ---
>
> [1] Hu, Yujing, et al. "Learning to utilize shaping rewards: A new approach of reward shaping." NeurIPS 2020.
>
> [2] Mguni, David, et al. "Learning to shape rewards using a game of two partners." AAAI 2023.
>
> [3] Wang, Yawei, and Xiu Li. "Reward function shape exploration in adversarial imitation learning: an empirical study." IEEE ICAICA 2021.
>
> [4] Zha, Daochen, et al. "Experience replay optimization." IJCAI 2019.
>
> [5] Narvekar, Sanmit, and Peter Stone. "Learning Curriculum Policies for Reinforcement Learning." AAMAS 2019.
>
> [6] Kostrikov, Ilya, et al. "Imitation Learning via Off-Policy Distribution Matching." ICLR 2020.
>
> [7] Kostrikov, Ilya, et al. "Discriminator-Actor-Critic: Addressing Sample Inefficiency and Reward Bias in Adversarial Imitation Learning." ICLR 2019.

---

### Official Review · Reviewer_n5K7 · 2025-10-29

**Soundness:** 3
**Presentation:** 3
**Contribution:** 3
**Rating:** 6
**Confidence:** 5

**Summary:**

This paper introduces ​​Reinforced Imitation Learning (RILe)​​, a novel framework that addresses the challenge of learning from expert demonstrations in high-dimensional environments. The core problem is that existing methods face a trade-off: ​​Inverse Reinforcement Learning (IRL)​​ provides dense, nuanced reward signals but is computationally expensive due to its iterative nature, while ​​Adversarial Imitation Learning (AIL)​​ methods like GAIL are efficient but often provide sparse, binary rewards that offer poor guidance.
RILe's key innovation is a ​​trainer-student framework​​ where both the reward function and the policy are learned simultaneously via reinforcement learning.

This setup allows the trainer to learn a dynamic, context-sensitive reward function that adapts to the student's current proficiency, combining the dense reward learning of IRL with the single-loop efficiency of AIL. Extensive experiments on MuJoCo and LocoMujoco benchmarks show that RILe achieves state-of-the-art performance, particularly in high-dimensional tasks.

**Strengths:**

​​1. The core idea of using an RL agent to learn a reward-generating policy is highly original and represents a paradigm shift from existing methods.

​​2. The paper is backed by an extensive set of experiments covering ablation studies, computational analysis, fairness comparisons, and performance evaluations on diverse benchmarks. The results are consistently strong.

​​3. By achieving high performance with significantly better computational efficiency than IRL methods, RILe offers a more practical and scalable solution for imitation learning in complex domains.

4. ​​The paper provides deep analysis beyond task performance, including visualizations of reward landscapes and dynamics, which offers compelling evidence for the proposed mechanism.

**Weaknesses:**

1. The paper acknowledges that training the three-component system (student, trainer, discriminator) introduces stability challenges. While strategies like freezing the trainer are mentioned (and detailed in Appendix B), the main text lacks a discussion on how sensitive the method is to hyperparameters related to this stability (e.g., the frequency of freezing). A more quantitative analysis of this sensitivity would be helpful.

2. ​​The comparison is excellent but could be even more compelling by including a "naive" baseline of training multiple independent RL agents with different seeds. This would help isolate the improvement due to the coordinated trainer-student learning versus simply the capacity to learn multiple policies.

3. ​​While the method is intuitively well-motivated and empirically validated, a more formal theoretical analysis of the convergence properties of the two-level RL system would further strengthen the contribution.

4. While the trainer-student architecture is intuitively reasonable and novel, there are works that also adopt similar teacher-student architecture to address the challenges in learning from demonstrations. For example, [1] uses the teacher-student architecture to facilitate the reward learning in Inverse RL. If the settings are the same, then baseline comparison might be necessary. Otherwise, discussion of the difference of the settings or ideas would be helpful to make this paper more comprehensive.

[1] Wan, Z., Wu, J., Yu, X., Zhang, C., Lei, M., An, B., & Tsang, I. (2025). FM-IRL: Flow-Matching for Reward Modeling and Policy Regularization in Reinforcement Learning. arXiv preprint arXiv:2510.09222.

**Questions:**

1. The strategies in Appendix B (freezing the trainer, different buffer sizes) are crucial for stable training. How sensitive is RILe's final performance to the specific timing of the trainer freeze or the ratio of buffer sizes? Did you observe a wide range of workable settings, or does it require precise tuning?

2. The learned reward policy \pi is a central artifact. Beyond the visualization in Fig. 4, did you analyze the behaviorof the trained trainer agent? For instance, does it learn to provide higher rewards for states that are "closer" to the expert's state distribution, effectively learning a potential function or a curriculum?

3. ​​The trainer's reward is currently based on the discriminator's output. Did you explore alternative reward signals for the trainer that might be less adversarial, such as a metric based on state/distribution matching (e.g., MMD, Wasserstein distance), and how that might affect the overall performance and stability?

4. There are similar works that utilize trainer-student architecture for learning from demonstrations, like FM-IRL ([1]). What is the key difference of the settings and the idea? If your settings are the same, these works should be incorporated as baselines for comparison. Otherwise, discussion or acknowledgement would be necessary.

[1] Wan, Z., Wu, J., Yu, X., Zhang, C., Lei, M., An, B., & Tsang, I. (2025). FM-IRL: Flow-Matching for Reward Modeling and Policy Regularization in Reinforcement Learning. arXiv preprint arXiv:2510.09222.


Glad to raise the scores if these concerns are addressed.

---

> ### Author Response · Authors · 2025-12-02
> **Response to Reviewer n5K7 (Part 1)**
>
> We thank the reviewer for their positive assessment of our work’s originality, consistently strong results, and deep analysis beyond performance, and for recognizing the  “paradigm shift” of our trainer-student framework. We particularly appreciate the suggestions to compare against naive ensembles and to clarify the relationship with FM-IRL. We have updated the paper to address these points.
>
> ---
>
> **1. Hyperparameter Sensitivity & Stability (W1, Q1):**
>
> We added a comprehensive **Trainer Design Study in Section 5.3 (Figure 5)** to address this concern. We analyzed six key parameters of the trainer agent: buffer size, discount factor, network capacity, update frequency, entropy weight, and freezing point. As shown in **Figure 5f**, delaying the freeze until convergence ($\sim$1M steps) yields the highest returns, though the method remains stable (does not collapse) with earlier freezing. Smaller buffers for the trainer generally perform better (**Figure 5a**), but RILe achieves high performance across a wide range of buffer sizes.
>
> As **Figure 5** shows, while tuning improves asymptotic performance, RILe learns functional policies and achieves high performance across a broad range of hyperparameter configurations.
>
> ---
>
> **2. Comparison with Naive Baselines (W2):**
>
> We added **Appendix A.2 (Comparison with Multi-Student GAIL)** to test whether the improvements are merely due to increased learning capacity. We trained **GAIL-2** (2 independent student agents) and **GAIL-5** (5 independent student agents) using different seeds but sharing a common discriminator. We report the best-performing policy from the ensemble. As **Table 4** shows, RILe outperforms the parameter-matched GAIL-2 and even the larger GAIL-5 ensemble. Interestingly, the naive ensembles slightly underperformed single-agent GAIL. We hypothesize this is because the shared discriminator faces a "mixture" of student policies, making it more robust and harder to fool, which makes learning harder for individual students. This confirms that RILe’s advantage comes from the coordinated adaptive shaping of the trainer, not just increased capacity.
>
> ---
>
> **3. Convergence Properties (W3):**
>
> We agree that a full convergence analysis of the two-level (trainer–student) RL system would significantly strengthen the theoretical side, but it is beyond the scope of this rebuttal. In this revision, we updated the paper for a clearer and more accessible theoretical story. In **Section 4 (Trainer Agent)**, we now explicitly contrast RILe with AIL. AIL uses a fixed transformation of the discriminator output ($r(s,a)=g(D_\phi(s,a))$), while RILe’s trainer optimizes a discounted RL objective whose optimal policy depends jointly on $D$, environment dynamics, and the evolving student policy. We state our hypothesis that, except in the degenerate case $\gamma=0$, this behavior cannot be represented as a static mapping $g(D)$. We then tie this directly to the empirical results in **Section 5.1**, where the static-transform **GAIL-Exp** baseline fails to match RILe, and the gamma ablation in **Section 5.3**, which shows that low gamma values drive RILe back toward GAIL-like performance, whereas a high discount factor is crucial for its gains. In **Appendix D**, we further discuss the difference between the trainer agent and a static transformation 3.We also explicitly mention a full convergence and equilibrium analysis as important future work in **Section 6 (Discussion)**.
>
> ---
>
> **4. Comparison with FM-IRL (W4, Q4)**
>
> We have added a discussion of FM-IRL [1] to **Section 2 (Related Work)**. FM-IRL uses a teacher-student architecture for density estimation (learning a stationary reward function via Flow Matching). In contrast, RILe uses a trainer for reward generation (learning a non-stationary policy via RL). While FM-IRL focuses on improving the accuracy of the static signal, RILe focuses on the delivery of the signal (the coach), optimizing a discounted objective to guide the student. Therefore they are orthogonal approaches; one could in principle use an FM-based evaluator from FM-IRL inside RILe.

---

> > ### Author Response · Authors · 2025-12-02
> > **Response to Reviewer n5K7 (Part 2)**
> >
> > **5. Learned Reward Policy & Curriculum (Q2):**
> >
> > We analyzed the trainer's behavior using three quantitative metrics alongside the visual landscape.
> >
> > * **Dynamic Potential Function (Sec 5.1 & 5.3):** Our static-transformation ablation (**Section 5.1**) suggests that the trainer does more than simply mapping “closeness to expert” to rewards. If the trainer were merely a static proximity detector, the GAIL-Exp baseline (which uses the exact same reward formula without a trainer policy) would match RILe’s performance. The fact that RILe outperforms it, combined with the gamma ablation (**Section 5.3**) showing the necessity of a high $\gamma$, suggests that the trainer learns a dynamic potential function that assigns value based on future expected success, not just current state similarity.
> >
> > * **Performance Correlation (Appendix A.3):** We measured the Correlation between Performance and Reward (CPR). We observe that increases in the learned reward align with gains in true environment performance (Positive CPR), whereas static baselines often decouple.
> >
> > * **Curriculum Mechanism:** Synthesizing these findings with the visualization in **Figure 4**, we observe that the trainer leverages its value function to generate a broad reward landscape early in training (guiding the student from random initialization). As the student improves, the reward concentrates around regions that most effectively guide the policy toward the expert path, creating an implicit curriculum based on the current stage of the student.
> >
> > ---
> >
> > **6. Alternative Reward Signals (MMD) (Q3):**
> >
> > We added **Section 5.1** to test whether we can use MMD as a discriminator replacement. We implemented **RILe-MMD**, replacing the adversarial discriminator with a Maximum Mean Discrepancy evaluator. RILe-MMD learns a functional policy but achieves lower returns than the adversarial version (**Figure 3**). This demonstrates RILe’s modularity (it works with non-adversarial signals), but suggests that for high-dimensional control, the discriminative signal provides a stronger gradient for the trainer. Further optimization in this direction might result in more robust variants of RILe, which we mention in the discussion.
> >
> > ---
> >
> > We thank the reviewer for their detailed and encouraging feedback, and hope that the new trainer-design analysis, multi-GAIL baseline, FM-IRL contextualization, and alternative evaluator experiment (RILe-MMD) address the reviewer's concerns.
> >
> > ---
> >
> > [1] Wan, Zhenglin, et al. "FM-IRL: Flow-Matching for Reward Modeling and Policy Regularization in Reinforcement Learning." arXiv preprint arXiv:2510.09222 (2025).

---

### Official Review · Reviewer_uDw8 · 2025-11-01

**Soundness:** 2
**Presentation:** 3
**Contribution:** 2
**Rating:** 2
**Confidence:** 4

**Summary:**

RILe frames imitation as a teacher–student game: a trainer RL policy shapes rewards from a discriminator while a student learns the control policy. The aim is IRL-like, adaptive feedback without IRL’s outer loop and to alleviate AIL’s near-binary/sparse signals. Strong numbers are reported on MuJoCo/LocoMujoco, with qualitative evidence that rewards adapt over training.

**Strengths:**

Clear, appealing reframing: optimizing a teacher policy instead of using a static discriminator reward.

Practical promise: dynamic reward shaping within a single loop; results are competitive/strong on high-dimensional control.

Modular: can plug in stronger discriminators and, in principle, other evaluators

**Weaknesses:**

Baseline oddity: In several tasks GAIL is the strongest/near-strongest baseline. For a 2016 method, this is atypical and raises concerns about coverage (missing stronger recent IL/IRL/offline-IL variants) and tuning fairness (architectures, budgets, entropy, normalization, replay).

Sparsity not fundamentally addressed: The trainer still derives its teaching signal from the discriminator; when the student is far from expert support, feedback can remain sparse/off-manifold. The method looks like reshaping/time-spreading discriminator feedback rather than densifying it (e.g., via expert-proximity surrogate rewards).

Complexity/stability: Joint training of student–trainer–discriminator increases tuning burden; sensitivity to freezes, entropy, buffer design is under-analyzed.

**Questions:**

Why is GAIL so strong here? Provide compute-normalized, architecture-matched tuning logs; if GAIL wins, diagnose (reward normalization, discriminator SNR, replay horizon, entropy, etc.).

Does RILe increase reward density when far from expert states? Report reward/advantage density conditioned on distance-to-expert (e.g., occupancy KL or NN-distance bins).

Ablate the trainer discount/entropy (incl. γ_T=0): when does the trainer collapse to a static transform of the discriminator?

Compare to densification baselines (e.g., expert-proximity surrogate rewards) and to stronger recent IL/IRL/offline-IL families (DICE/ValueDICE, DAC, IQL/CQL-fD), under matched budgets.

---

> ### Author Response · Authors · 2025-12-02
> **Response to Reviewer uDw8**
>
> We thank the reviewer for their detailed assessment and for acknowledging the “clear, appealing reframing” and “competitive/strong” results of our method. We added additional experiments and analyses to address the reviewer’s concerns regarding baselines, reward density, and trainer design.
>
> ---
>
> **1. Baseline Coverage and GAIL Strength (W1, Q1, Q4):**
>
> * **ValueDICE & DAC:** We included **ValueDICE** [1] and **DAC** [2] in our efficiency vs. performance analysis in **Section 5.4 (Figure 6)**. While these off-policy methods improve sample efficiency, RILe still achieves superior asymptotic performance in high-dimensional tasks like *HumanoidStand*.
>
> * **TDIL (expert-proximity surrogate rewards)**: We added **TDIL** [3] as a densification baseline. We compare RILe with it in both the reward landscape analysis (**Section 5.2, Figure 4**) and robustness study (**Section 5.6, Table 2**). As Figure 4 shows, even though TDIL generates dense rewards, they are fairly *static*. In contrast, RILe changes the reward landscape significantly to adapt to the current stage of the student.
>
> * **Fairness:** All methods use identical architectures and budgets. The "strength" of GAIL in our results stems from using an optimized implementation (SAC with entropy regularization). However, as shown in **Figure 6**, GAIL's performance plateaus significantly earlier than RILe's in complex environments.
>
> ---
>
> **2. Reward Density and "Off-Manifold" Guidance (W2, Q2):**
>
> * **Static vs. Adaptive Density:** We visualize reward landscapes in **Section 5.2 (Figure 4)** to empirically show how RILe’s reward function adapts to the student. The visualization shows that GAIL and AIRL produce rewards concentrated narrowly around the expert path (sparse). While TDIL rewards are denser, they remain relatively static during training, guiding the agent to local maxima. In contrast, RILe’s trainer learns a shaped, dense landscape, which provides meaningful rewards to guide the student back to the manifold when the student is far away from the expert trajectory.
>
> * **Off-Manifold Guidance:** Crucially, RILe’s reward function is dynamic. Early in training, non-zero rewards cover a broad region to guide the student from random initialization (off-manifold) toward the expert. As the student improves, the reward concentrates around regions that most effectively guide the policy toward the expert path. This *curriculum of density* explains why RILe succeeds where static discriminators vanish.
>
> ---
>
> **3. Trainer Gamma/Entropy Ablation (Q3):**
>
> We added two new analyses to compare the trainer with static transformations or degenerate settings ($\gamma = 0$).
>
> * **Gamma Ablation (Section 5.3):** We study the effect of the trainer’s discount factor (**Figure 5b**). Results show that low $\gamma$ values lead to performance degradation down toward GAIL-like scores, while a higher $\gamma$ is critical for RILe’s peak performance. This supports the view that the trainer leverages long-horizon effects rather than only immediate discriminator signals when guiding the student.
>
> * **Static Transform Comparison (Section 5.1):** We explicitly tested if the trainer is just a reshaping function. We trained a GAIL agent using the exact reward transformation used by our trainer's objective ($r(s,a)=e^{-|1-(2D_{\phi}(s,a)-1)|}$). This static transformation (**GAIL-Exp**) fails to match RILe's performance (**Figure 3**). This empirically shows that the active, non-myopic optimization of the trainer policy provides value that a static transformation function cannot.
>
> ---
>
> **4. Stability Analysis (W3):**
>
> We added a comprehensive hyperparameter study in **Section 5.3 (Figure 5)** analyzing six key trainer parameters, including buffer size, update frequency, and freezing point. The results demonstrate that while RILe benefits from tuning, it remains stable across a reasonably wide range of settings (e.g., high performance across various network capacities and entropy weights).
>
> ---
>
> We believe these new experiments address the reviewer’s concerns about baselines, reward density, and trainer complexity/stability.
>
> ---
>
> [1] Kostrikov, Ilya, Ofir Nachum, and Jonathan Tompson. "Imitation Learning via Off-Policy Distribution Matching." International Conference on Learning Representations. 2020.
>
> [2] Kostrikov, Ilya, et al. "Discriminator-Actor-Critic: Addressing Sample Inefficiency and Reward Bias in Adversarial Imitation Learning." International Conference on Learning Representations. 2019.
>
> [3] Chiang, Chia-Cheng, et al. "Expert proximity as surrogate rewards for single demonstration imitation learning." Proceedings of the 41st International Conference on Machine Learning. 2024.

---

### Author Response · Authors · 2025-12-02
**General Response - Summary of Strengths & Rebuttal Revisions**

Dear AC and Reviewers,

We provide this summary to assist the evaluation of the paper. We first highlight the **consensus strengths** identified by the reviewers, followed by a summary of the **key concerns and new experiments**. We thank all the reviewers for their time and insightful feedback.

---

**1. Consensus Strengths: Why Reviewers Value RILe**

Despite initial concerns, reviewers were unanimous in recognizing the novelty and performance of RILe:

* **Originality & Paradigm Shift:** Reviewer **n5K7** called the core idea `highly original` and a `paradigm shift from existing methods`. **uDw8** praised the `clear, appealing reframing` and **NWQg** noted the approach is `original, introducing a novel paradigm` and `meaningfully advances the field by addressing long-standing issues of myopic reward learning`.

* **Extensive Analysis & Motivation:** Reviewer **n5K7** emphasized that the paper is `backed by an extensive set of experiments` and `provides deep analysis beyond task performance, which offers compelling evidence`. **NWQg** stated the `technical quality is strong, with clear theoretical motivation` while **Ysyr** noted there are `good number of experiments exploring the properties of various components`.

* **Strong Performance:** Reviewers consistently acknowledged strong results. **uDw8** noted `results are competitive/strong on high-dimensional control`. **n5K7** described the results as `consistently strong` and `state-of-the-art`. **NWQg** highlighted `comprehensive experiments demonstrating consistent improvements in performance, robustness, and efficiency` and **Ysyr** identified our robustness results as `pretty strong and interesting`.

---

**2. Addressing Key Concerns: New Evidence & Revisions**

We have addressed the primary concerns with a set of new and extended studies in the revised PDF:

**A. Trainer Necessity: Active Optimization vs. Static Transformation (uDw8, Ysyr, NWQg)**

* *Critique:* Is the trainer simply learning a static mathematical transformation?

* *New Study (Sec. 5.1):* We compared RILe against **GAIL-Exp** (uses the exact static reward transformation of our trainer) and **GAIL-Scaled**. RILe significantly outperforms both  (**Fig. 3**), empirically suggesting the performance gain stems from the trainer's **non-myopic RL optimization** rather than static transformation.

* *New Study (Sec. 5.3):* We show that high $\gamma$ values are required for peak performance; with lower $\gamma$, RILe’s performance degrades to GAIL levels, confirming the trainer leverages long-horizon values.

**B. Stability & Trainer Design (n5K7, uDw8, NWQg)**

* *Critique:* How sensitive is RILe to trainer hyperparameters?

* *New Study (Sec. 5.3):* We added a new study **Fig. 5** ablating six key trainer parameters. RILe achieves high performance across broad configurations (e.g., varying capacity, entropy, gamma), showing it is not brittle.

**C. Learning Capacity & Signals (n5K7, uDw8)**

* *Critique:* Do gains come simply from increased capacity? Could simpler signals (MMD) replace the discriminator?

* *New Study (App. A.2)*: We trained **Naive Ensembles** (GAIL-2/5). RILe outperforms even the 5-agent ensemble, suggesting the trainer-student adaptation provides value beyond capacity.

* *New Study (Sec. 5.1):*  We tested **RILe-MMD** (replacing discriminator with MMD). While functional (showing modularity), it underperforms the discriminator-based RILe.

**D. Baseline Coverage (uDw8, Ysyr)**

* *Critique:* Need comparison to recent off-policy and robust/densification methods.

* *Extended Sec. 5.4:* We added **ValueDICE** and **DAC** as off-policy methods. RILe achieves higher asymptotic performance than both while maintaining computational efficiency.

* *Extended Sec. 5.2 and 5.6*: We added **TDIL** to robustness/reward landscape experiments as a robust densification baseline. RILe matches the robustness of TDIL under action noise while outperforming TDIL in clean/state-noise settings. RILe also creates a dynamic curriculum for the student, whereas the TDIL’s reward remains dense but static.

**E. Theoretical Clarity (NWQg, Ysyr, n5K7)**

* *Critique:* Provide better intuition on why the trainer is not a static map.

* *Extended Sec. 4:* We refined the text to explicitly state our intuition on the trainer's discounted objective against the myopic static map of AIL .We link this intuition to the results in Sec. 5.1 and 5.3, and the detailed discussion in App. D.

**F. Extended Related Work (Ysyr, n5K7)**

* *Extended Sec. 2:* We contextualized RILe against FM-IRL (*stationary density estimation* vs. *dynamic reward generation*); reward shaping and curriculum learning approaches.

---

**Conclusion**

With the inclusion of the static-transform ablation, stability analysis, and robust baselines, we believe we have rigorously addressed the main concerns. The revised paper presents a framework that offers the efficiency of AIL with the granular, adaptive guidance of IRL.

---

### Meta-Review · Area_Chair_Cxvb · 2025-12-28

**Summary:**

My recommendation for this submission is Reject. While several reviewers highlighted the extensive experimental results provided in the paper, most reviewers also had concern on the stability and convergence guarantee of the proposed framework that introduces two RL agents. The rebuttal have provided the authors' intuition on this concern, unfortunately it is not what the reviewers requested. While I don't expect every conference paper to have a theoretical foundation, considering the complex nature of the proposed framework, I agree with the reviewers in that the paper needs a formal analysis on the convergence. It's quite important for the new framework to have such a guarantee, because that would motivate the community to have confidence in the direction -- and put more effort on *making it work* by further stabilizing and improving the practical implementation, like various follow-up work to GAIL improved it while not fundamentally touching the objective itself. I would like to encourage authors to incorporate these reviewers' points into the revision.

**Reviewer Concerns:**

Concerns addressed by the rebuttal
- Strong performance of GAIL
- Concern on the reward density/sparisty claim
- More baselines
- Comparison to a baseline that uses the transformed discriminator reward
- More discussion on relevant work
- Cost analysis

Outstanding concerns
- Stability challenges & Theoretical analysis on the convergence: While the authors have provided additional stability analysis (which is sensitivity analysis to hyperparameters), it didn't fully resolve the concern on the stability challenges of the proposed framework. Several reviewers requested theoretical analysis on this point but the rebuttal didn't provide this.

**Reviewer Scores:**

- Reviewer uDw8 (score 2): I expect this reviewer to increase their score to borderline score considering that most of the concerns are resolved, but not to champion the paper.
- Reviewer n5K7 (score 6): I expect this reviewer to remain their score and not to champion the paper considering that some of concerns (e.g., formal theoretical analysis) remain.
- Reviewer Ysyr (score 4): I expect this reviewer to remain their negative score considering that their main concern on the soundness of the method still remains.
- Reviewer NWQg (score 4): I expect this reviewer to maintain their score or update the score to borderline considering that the main concerns are at least partially resolved.

---

### Decision · Program_Chairs · 2026-01-26

Reject